# The *Toxoplasma* monocarboxylate transporters are involved in the metabolism within the apicoplast and are linked to parasite survival

Hui Dong[1,2†], Jiong Yang[3†], Kai He[1,2†], Wen-Bin Zheng[4], De-Hua Lai[3], Jing Liu[1,2], Hui-Yong Ding[1,2], Rui-Bin Wu[1,2], Kevin M Brown[5], Geoff Hide[6], Zhao-Rong Lun[3], Xing-Quan Zhu[4], Shaojun Long[1,2*]

[1]National Key Laboratory of Veterinary Public Health Safety, and College of Veterinary Medicine, China Agricultural University, Beijing, China; [2]National Animal Protozoa Laboratory and School of Veterinary Medicine, China Agricultural University, Beijing, China; [3]MOE Key Laboratory of Gene Function and Regulation, State Key Laboratory of Biocontrol and Guangdong Provincial Key Laboratory of Aquatic Economic Animals, School of Life Sciences, Sun Yat-Sen University, Guangzhou, China; [4]College of Veterinary Medicine, Shanxi Agricultural University, Taigu, China; [5]Department of Microbiology and Immunology, University of Oklahoma Health Sciences Center, Oklahoma City, United States; [6]Biomedical Research and Innovation Centre and Environmental Research and Innovation Centre, School of Science, Engineering and Environment, University of Salford, Salford, United Kingdom

**\*For correspondence:**
LongS2018@163.com

[†]These authors contributed equally to this work

**Competing interest:** The authors declare that no competing interests exist.

**Abstract** The apicoplast is a four-membrane plastid found in the apicomplexans, which harbors biosynthesis and organelle housekeeping activities in the matrix. However, the mechanism driving the flux of metabolites, in and out, remains unknown. Here, we used TurboID and genome engineering to identify apicoplast transporters in *Toxoplasma gondii*. Among the many novel transporters, we show that one pair of apicomplexan monocarboxylate transporters (AMTs) appears to have evolved from a putative host cell that engulfed a red alga. Protein depletion showed that AMT1 and AMT2 are critical for parasite growth. Metabolite analyses supported the notion that AMT1 and AMT2 are associated with biosynthesis of isoprenoids and fatty acids. However, stronger phenotypic defects were observed for AMT2, including in the inability to establish *T. gondii* parasite virulence in mice. This study clarifies, significantly, the mystery of apicoplast transporter composition and reveals the importance of the pair of AMTs in maintaining the apicoplast activity in apicomplexans.

## eLife assessment

This study identifies two new transporters in the apicoplast, a non-photosynthetic organelle of apicomplexan parasites. While this is **important** work, it only partially reveals how essential these transporters are, as it does not address the metabolic function of the transporters for the parasite. Although the evidence is still **incomplete**, the results should be of interest to parasitologists and eukaryotic cell biologists.

## Introduction

The phylum Apicomplexa encompasses thousands of intracellular parasites, including *Plasmodium* spp., the notorious agent of malaria, and *Toxoplasma gondii*, the highly successful parasite that infects almost all warm-blooded animals and humans (*Plattner and Soldati-Favre, 2008*). It is believed that apicomplexan evolution included two key independent, and sequential, endosymbiotic events, finally leading to the common ancestor of the chromalveolates (e.g. alveolates, heterokonts, oomycetes, and Cryptomonads) (*Cavalier-Smith, 1999*; *Gould et al., 2008*; *Janouskovec et al., 2010*). This ancestor eventually gave rise to the apicomplexans, most of which still contain a secondary non-photosynthetic plastid, the apicoplast (*McFadden and Yeh, 2017*; *Oborník et al., 2009*; *Salomaki and Kolisko, 2019*). Though endosymbiont origins of the apicoplast were proposed (*Köhler et al., 1997*; *McFadden et al., 1996*; *Wilson et al., 1996*), it is believed that the apicoplast originates from a red alga (*Foth and McFadden, 2003*; *McFadden and van Dooren, 2004*). Therefore, the apicoplast membranes have been proposed to evolve from the red algal plastid, the algal plasma membrane, and the host endomembrane when counting from the innermost to outermost membranes of the apicoplast (*Gould et al., 2008*; *McFadden and Yeh, 2017*). This heavily compartmentalized feature poses great challenges for metabolic integration of the organelle within these parasites and, in particular, the transport of metabolites across these membrane systems.

The apicoplast houses essential biosynthetic pathways. These pathways include the fatty acid synthesis type II (FASII) pathway, methylerythritol phosphate (MEP) pathway, Fe–S cluster biosynthesis pathway, (partial) heme biosynthesis pathway, and lysophosphatidic acid synthesis pathway (using the FASII products) (*Amiar et al., 2016*; *Fata and Rabuzzi, 1988*; *Jomaa et al., 1999*; *Kloehn et al., 2021*; *Ralph et al., 2004*; *Tjhin et al., 2020*; *Waller et al., 1998*). They are able to make critical components of fatty acids and isoprenoid precursors (IPP) for the parasites, in *T. gondii* and *Plasmodium falciparum* (*McFadden and Yeh, 2017*; *Ramakrishnan et al., 2012*; *Striepen, 2011*; *Vaughan et al., 2009*; *Yeh and DeRisi, 2011*). The apicoplast FASII pathway in *T. gondii* is key to understand the metabolic flexibility, and can become essential under metabolic adaptation conditions (*Amiar et al., 2020*; *Botté et al., 2013*; *Krishnan et al., 2020*; *Primo et al., 2021*). Moreover, the apicoplast maintains housekeeping activities for genome replication, transcription, and translation (*Low et al., 2018*; *McFadden and Roos, 1999*). Accordingly, it requires adequate supplies of carbon molecules, energy, reducing equivalents, cofactors, amino acids, nucleotides as well as inorganic ions, to feed the biosynthetic pathways and plastid maintenance activities in the matrix (*Kloehn et al., 2021*; *McFadden and Yeh, 2017*; *Ralph et al., 2004*). However, compared to plant plastids, the nature of the channels and transporters in the apicoplast remains obscure.

The apicoplast biosynthetic pathways were characterized through a homology/bipartite signal-based approach (*Foth et al., 2003*; *Ralph et al., 2004*). A similar approach was utilized to identify phosphate translocators in *T. gondii* (i.e. *Tg*APT1) and *P. falciparum* (i.e. *Pfi*TPT and *Pfo*TPT), and a two-pore channel TPC in *T. gondii*. These transporters are capable of supplying phosphorylated carbons (*Brooks et al., 2010*; *Fleige et al., 2007*; *Karnataki et al., 2007*; *Lim et al., 2010*; *Mullin et al., 2006*), and calcium to the apicoplast (*Li et al., 2021*). It was proposed that an early step in the endosymbiotic event was to integrate transporters that exchange metabolites across the membranes (*Cavalier-Smith, 2000*; *Weber et al., 2006*). Consistent with that, studies demonstrated that 57–58% of the chloroplast transporters have an origin in the putative host cell in the first endosymbiosis while 20–30% are from bacteria (*Karkar et al., 2015*; *Tyra et al., 2007*). It is plausible that a complex transporting system was produced in the evolution of the secondary plastid (i.e. the apicoplast).

Recent studies suggested that we still do not know the major source of pyruvate required for the biosynthesis pathways in the apicoplast of *T. gondii* and *P. falciparum* (*Swift et al., 2020*; *Xia et al., 2019*). Therefore, identifying the apicoplast transporter composition and their functions is critical for an understanding of how these transporters evolved and how the organelle maintains its metabolic and housekeeping activities. In this study, we utilized the apicomplexan model organism *T. gondii* to screen apicoplast membrane proteins, identified from available computational and proteomic information (*Barylyuk et al., 2020*; *Boucher et al., 2018*) and from our APT1-TurboID proximity proteomic candidates, and discovered several novel apicoplast transporters. Among the many transporter candidates, one pair of monocarboxylate transporters (AMTs) appeared to have evolved from the putative host cell involved in the evolution of the secondary plastid. Detailed analyses support association of the AMTs with the apicoplast biosynthesis pathways likely by supplying the pathways with substrates.

This study opens up the possibilities of tackling difficult questions of substrate transport and especially for the pair of monocarboxylate transporter (MCT) in *T. gondii* and in other members of this phylum of deadly parasites.

## Results

### Discovery of novel transporters in the apicoplast of *T. gondii*

To identify apicoplast transporter proteins in *T. gondii*, we first analyzed candidates identified in recent studies using combined proteomic and computational analyses in *T. gondii* and *P. falciparum* (*Barylyuk et al., 2020*; *Boucher et al., 2018*). The hyperplexed localization of organelle proteins (hyperLOPIT) assigned 129 proteins to the apicoplast (*Barylyuk et al., 2020*), of which three proteins were likely transporters, that is APT1 (transmembrane domain, TMD = 6), one major facilitator superfamily (MFS) transporter (TGME49_309580) (HAP1, hypothetical apicoplast protein 1), and one hypothetical protein (TGME49_267690) (HAP2) (*Supplementary file 1a*). We also mined an apicoplast proteomic dataset obtained by the BioID fusion at acyl carrier protein (ACP) in *P. falciparum* (*Boucher et al., 2018*), which reported 16 TMDs ≥6 protein candidates, and two additional candidates annotated as ABC transporters (TMDs <6). We identified the orthologs in *T. gondii* using the *P. falciparum* candidates (*Boucher et al., 2018*; *Sayers et al., 2018*), resulting in eight candidates in *T. gondii*. They were temporarily named as HAP3–9 (*Supplementary file 1a*). Taken together, these datasets provided only limited numbers of candidate transporters.

To better define the repertoire of apicoplast transporters, we employed a biotin proximity labeling strategy that is useful for defining organelle proteomes. A promiscuous biotin ligase tag called TurboID was endogenously fused with APT1, using a CRISPR approach (*Supplementary file 1b-d*; *Branon et al., 2018*; *Brooks et al., 2010*; *Long et al., 2018*). The apicoplast acetyl-CoA carboxylase 1 (ACC1) is a strongly biotinylated protein (*Zuther et al., 1999*), which might provide a clear and strong streptavidin labeling at the apicoplast (*Figure 1A*). Though the streptavidin signal and the protein itself were slightly diffused (*Karnataki et al., 2007*), APT1 was reported to mainly localize to four of the apicoplast membranes by an immuno-electron microscopy (immuno-EM) approach for HA epitopes tagged at the C-terminus of APT1 (*Karnataki et al., 2007*). This feature of APT1 thus provided good biotinylation labeling at potential proteins localized to membranes of the organelle. Predictably, many additional bands present on the APT1-TurboID lane were likely to be apicoplast proteins (*Figure 1B*). The resultant mass-spectrometry datasets (N = 3 for each) (*Supplementary file 2a*) were analyzed by comparing the APT1 fusion to the parental line (*Figure 1C* and *Supplementary file 2b*). This analysis showed that APT1 itself has the highest p-value associated with its enrichment in the TurboID line, as would be expected for a bait protein. The TMD candidates (TMD ≥6 and 2 ≤ TMD < 6) were evenly distributed on the volcano map plotted by fold changes against p-values in a similar pattern to the distribution of known apicoplast proteins (*Figure 1C* and *Supplementary file 2c*). These features suggested that APT1-TurboID had enriched apicoplast proteins. Thereafter, we compiled a list of the hits in order of TMD numbers, and selected all transporter candidates (including carriers) and hypothetical proteins (TMD ≥6) as candidates (*Supplementary file 2d*). These candidates were merged with the previously described ones (HAP1–9) and temporarily named as HAP10–50 (*Supplementary file 2e*).

After the above candidate selections, as described in our workflow (*Figure 1D*), we compiled a list of 50 candidates (HAP1–50) for endogenous tagging and IFA analysis in *T. gondii*. We successfully generated endogenous HAP-6Ty fusion lines (46 HAPs) using a CRISPR-Cas9 tagging approach (*Figure 1—figure supplement 1A* and *Supplementary file 2e*), as described in our previous study (*Long et al., 2017b*). Indirect fluorescence assay (IFA) showed that many HAP-6Ty fusions were localized to foci inside the parasites, resembling the position of the apicoplast location (*Figure 1—figure supplement 1A*), whereas some candidates were likely to be in the cytosolic vesicles (e.g. HAP13, HAP30, and HAP35), the apical region (e.g. HAP19), or other structures (e.g. HAP22) (*Figure 1—figure supplement 1B*). Further analysis was performed by co-localization of the fusions with an apicoplast marker ACP, which showed that many of the foci had only partial or no co-localization with ACP (*Figure 1—figure supplement 2A*). However, strict analyses clearly identified 15 HAP fusions that were almost perfectly co-localized with ACP (*Figure 1E*).

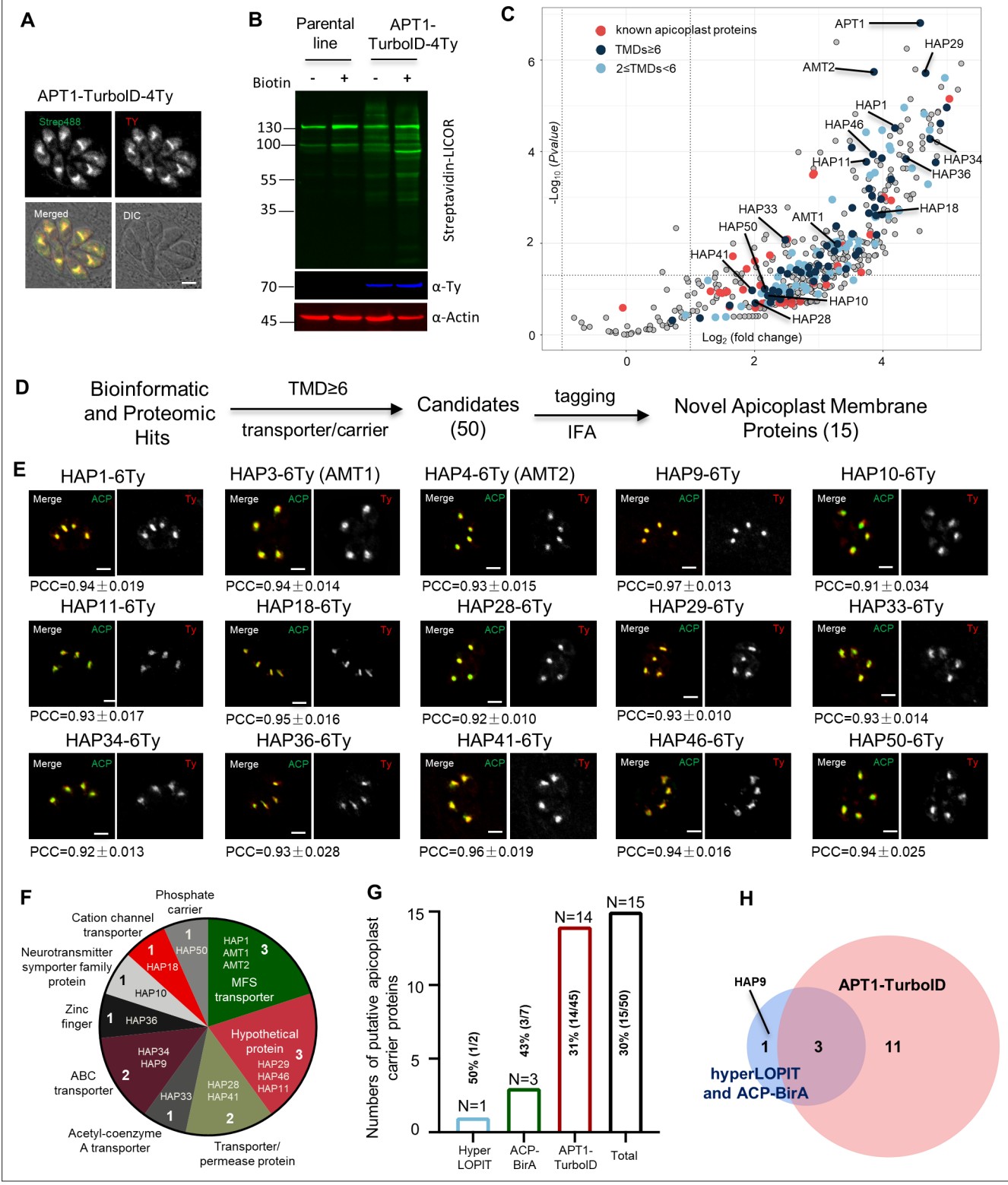

**Figure 1.** Discovery of apicoplast membrane proteins. Activity test of biotinylation of proximal proteins by APT1-TurboID-4Ty. Parasites were grown in 500 µM D-biotin for 90 min, followed by detection of biotinylated proteins (green) by streptavidin reagents on indirect immunofluorescence assay (IFA) (**A**) and on western blots (**B**). Actin served as the loading control. Scale = 5 µm. (**C**) Volcano plot analysis comparing the TurboID fusion to the parental line. Three replicate mass-spectrometry experiments were analyzed by the Student *t*-test. Known apicoplast proteins (red) and candidates with different numbers of transmembrane domains (TMDs; deep and light blue were indicated, and novel apicoplast proteins were pointed out; see ***Supplementary***

*Figure 1 continued on next page*

*Figure 1 continued*

**file 2a**). (**D**) Workflow of the discovery of novel apicoplast membrane proteins. Candidates from hyperLOPIT (**Barylyuk et al., 2020**), ACP-BirA (**Boucher et al., 2018**), and APT1-TurboID (this study) were screened by epitope tagging and IFA. (**E**) Confocal co-localization of HAP-6Ty with the apicoplast marker ACP, showing confocal imaging of HAP-6Ty (red) and ACP (green), and Pearson correlation coefficiency (PCC). The PCC values over the merged fluoresent foci for candidate proteins and ACP were analyzed by a co-localization and region of interest (ROI) intensity analysis module in the NIS Elements AR software and shown with mean ± standard error of the mean (SEM; *N* = 6 for each). Scale = 5 μm. (**F–H**) Summary and comparison of the screening from three datasets. The putative transporters are grouped into diverse types (**F**) and identified from three different datasets (**G**), which shared three novel proteins, as shown by the Venn diagram (**H**). Three independent experiments were performed with similar outcomes (**A, B, E**).

The online version of this article includes the following source data and figure supplement(s) for figure 1:

**Source data 1.** PDF file containing the uncropped western blot gels for verification of the TurboID fusion lines in *Figure 1A*.

**Source data 2.** Excel file containing the raw images of confocal analysis for verification of co-localization of newly discovered proteins with the apicoplast marker ACP.

**Figure supplement 1.** Indirect fluorescence assay (IFA) analysis of HAP-6Ty fusions in parasites.

**Figure supplement 2.** Analyses of non-apicoplast-localized HAP fusions, and domain analysis of novel apicoplast transporters.

Of the 15 novel apicoplast membrane proteins, we identified in *T. gondii* that 12 proteins are annotated as putative transporters or carriers in TOXODB, and 3 are annotated as hypothetical proteins (TMD ≥6) (*Figure 1F* and *Supplementary file 2f*). These novel proteins included 1 from hyperLOPIT, 3 from ACP-BirA, and 14 from our APT1-TurboID (*Figure 1G*), among which three of these proteins were shared between the APT1-TurboID and the combinations of hyperLOPIT and ACP-BirA (*Figure 1H*). Collectively, TurboID proved to be a reasonably robust approach for the identification of putative apicoplast transporters.

## Novel transporters are mostly conserved in the Apicomplexa

To determine if the apicoplast transporters we identified in *T. gondii* are conserved in the Apicomplexa, we searched for orthologs of these novel proteins in different species, as shown in *Figure 2A, B*. The chromalveolate hypothesis proposed a plastid origin in a common ancestor of a single red alga, which unites the Alveolates with the chromists (*Cavalier-Smith, 1999*). We identified orthologs in species belonging to the taxa using a powerful Hidden Markov Model (HMM) profiling strategy. The best hits (the cutoff *E*-value: $1 \times 10^{-7}$) remained on the list of resultant orthologs in these species, which were then used for conservation analyses in the ComplexHeatmap package in R (v4.2.0) (*Gu et al., 2016*; *Supplementary file 3a*, contains the hits with *E*-values). The conservation heatmap showed that proteins with conserved transport domains (HAP1, 3, 4, 9, 10, 28, 33, 34, 41, and 50) are mostly conserved in the Apicomplexa (*Figure 2A*), including the Gregarina and *Cryptosporidium* that have lost the apicoplast. Further analysis revealed that most orthologs with the exception of HAP3 and HAP4, can be identified in green algae, red algae, and plants (*Figure 2B*). As the red alga is considered to have given rise to the ancestor of the secondary plastid (*Gould et al., 2008*; *Janouskovec et al., 2010*), the results thus indicated that those transporters were possibly derived from the red alga. However, further analysis would be required to clarify their origins, as shown for chloroplastid transporters (*Karkar et al., 2015*).

## AMT1 and AMT2 might have critical roles in metabolic integration of the apicoplast

Several transporters, such as HAP1, 3, 4, and 34 and 50, are predicted to be essential in *T. gondii* and in *P. falciparum* (*Figure 2A, B*, and *Supplementary file 3a* contains the *P. falciparum* orthologs with fitness scores) (*Sidik et al., 2016*; *Zhang et al., 2018*). The analyses with InterProScan showed that three of these transporters (HAP1, 3, and 4) belong to the MFS (*Figure 1—figure supplement 2B*). The MFS is a large family of transporters capable of transporting small solutes in response to chemiosmotic ion gradients (*Pao et al., 1998*; *Walmsley et al., 1998*). These MFS transporters contain a pair of MCT with 11–12 TMDs (*Figure 1—figure supplement 2C*). The MCTs play essential roles in the transport of nutrients, such as pyruvate, lactate, amino acids and so on, in the plasma membrane of human cells (*Felmlee et al., 2020*). These features attracted our intense attention, and thus named the MCTs as **A**picoplast **M**onocarboxylate **T**ransporters (AMTs, AMT1 for HAP3, TGGT1_233540; AMT2 for HAP4, TGGT1_297245).

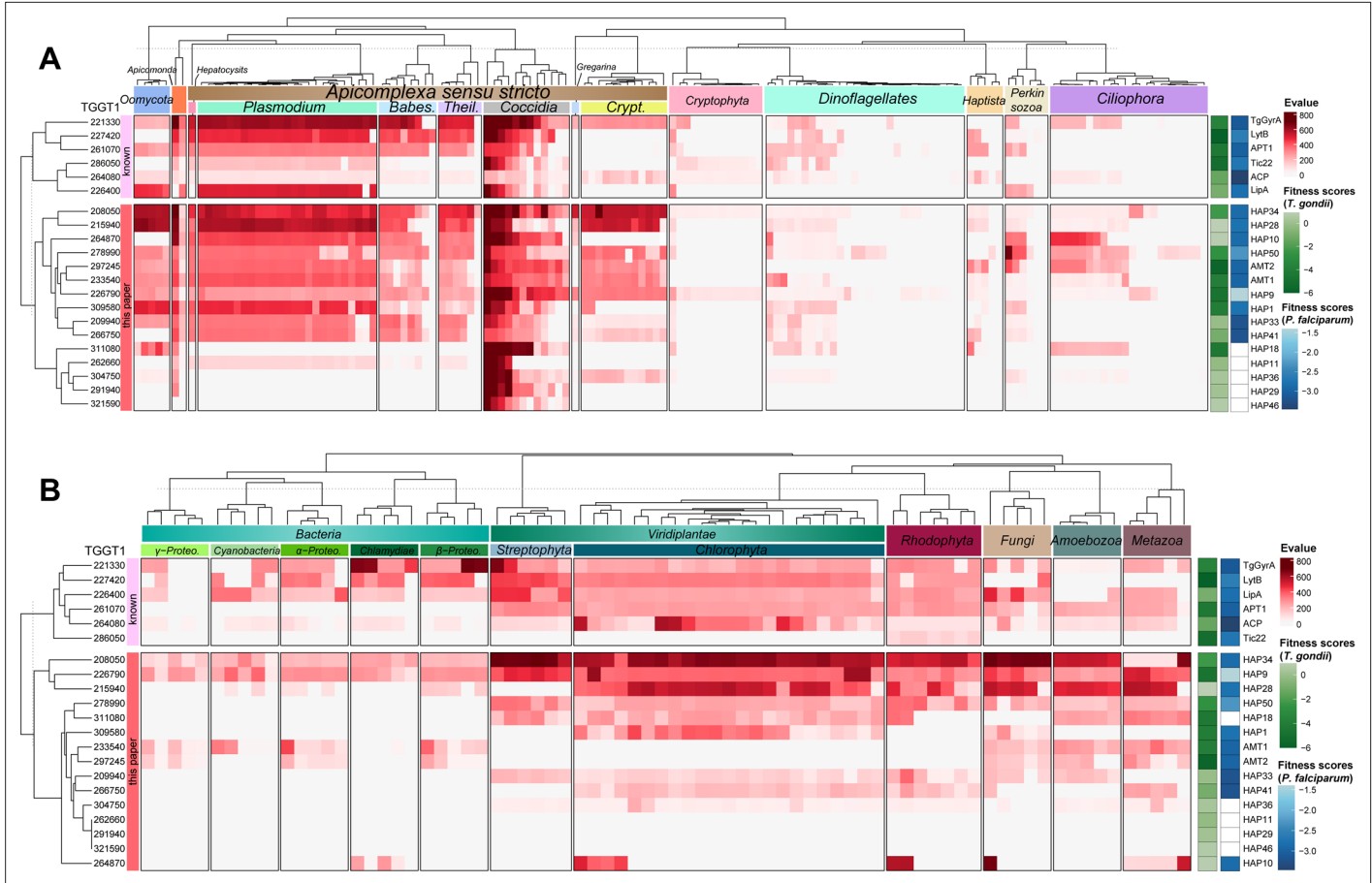

**Figure 2.** Conservation analysis of putative transporters. Heatmap indicating conservation of known apicoplast proteins and newly identified proteins among chromalveolates (**A**), red alga, green alga, plants, fungi, metazoa, and bacteria (**B**). The conservation was scaled by a heat bar (red) with *E*-values for putative orthologs of the listed proteins (see ***Supplementary file 3a***). ToxoDB protein numbers (left) and protein (temporary) names (right) are shown. Their sequences were searched for orthologs using the Hidden Markov Model (HMM) (*E*-value <*E*−7) and the best ortholog was used for the heatmap analysis in ComplexHeatmap package in R. Parasite fitness scores were shown on the right with the green heat bar for *T. gondii* (ToxoDB) (***Sidik et al., 2016***) and the blue bar for *P. falciparum* (PlasmoDB) (***Zhang et al., 2018***). The tree dendrogram across the genes were clustered according to the euclidean distance using UPGMA method. The species tree (top) shows phylogenetic relationships and major clades: *Theil.*, *Theileria*; *Babes*, *Babesia*; *Crypt.*, *Cryptosporidium*; *Proteo, Proteobacteria*.

We noticed that these two transporters are absent in red algae, green algae, and plants (***Figure 2B***). This observation is in contrast to APT1, which was identifiable in these primary plastid-containing taxa (e.g. plants) but not in cyanobacteria (***Figure 2B***). Evolution and phylogenetic analyses suggested that the plant orthologs of APT1 are derived from a putative host cell that previously engulfed a cyano-bacteria (***Knappe et al., 2003***; ***Weber et al., 2006***). The pair of AMTs appears similar to the APT1 situation, indicating that they were possibly derived from the heterotrophic protist – the putative host cell – that previously engulfed a red alga. The AMTs were identifiable in the photosynthetic relative (e.g. *Chromera velia*) of the apicomplexans, and a distant relative taxa of Oomycetes (***Figure 2A***), indicating a broader conservation in these species. We further analyzed orthologs in selected species of apicomplexans and found that most species contain at least two identifiable orthologs, including *Crytosporidium* spp. (***Supplementary file 3b***). Collectively, the pair of AMTs is likely to have critical roles in metabolic integration of the endosymbiont in the parasite cytosol during evolution.

## AMT1 and AMT2 are essential for the parasites

The confocal imaging showed that AMT1-6Ty was almost perfectly co-localized with AMT2-6HA (***Figure 3—figure supplement 1A***), suggesting that the putative transporters are highly likely to be localized at the apicoplast membranes. In support of this model, immuno-EM provided a further piece

of evidence, supporting the membrane localization of AMT1 and AMT2 (*Figure 3—figure supplement 1B*). Due to the complexity of the apicoplast membranes, and heterogeneity of immunolabeling, the precise topology of AMT1 and AMT2 within the apicoplast membranes merits future investigation. To analyze the physiological functions of the AMTs, we generated the conditional knockdown lines using an IAA-inducible degron (AID) system in *T. gondii* (*Brown et al., 2017*; *Long et al., 2017b*). The CRISPR-Cas9 genome editing approach was utilized to specifically fuse the AID-6Ty at the 3'-terminus of the genes in the IAA receptor (TIR1) expressing line RH TIR1 (*Figure 3—figure supplement 2*). To conditionally study both proteins together, we fused AMT2 with AID-3HA at the 3'-terminus in the background of AMT1-AID-6Ty, creating a double knockdown line AMT1-AID-6Ty/AMT2-AID-3HA (dKD). Correct integration of the AID fragments was confirmed by diagnostic polymerase chain reaction (PCR) as demonstrated in the schematic diagram (*Figure 3—figure supplement 2A, B*). As expected, the AID fusions were efficiently and stably depleted in IAA treatments for 3–36 hr, as shown on western blots (*Figure 3A*). It was noteworthy that western blots detected two bands in the single and double AID fusion lines, which might be the precursor and mature forms of AMT1 and AMT2 (*Figure 3A*). So far, several membrane proteins have been identified in the apicoplast of *T. gondii*, four of which were observed to be processed (FtsH1, TiC20, ATrx1, and TPC) (*DeRocher et al., 2008*; *Karnataki et al., 2007*; *Karnataki et al., 2009*; *van Dooren et al., 2008*). Yet, at least ATrx1 and FtsH1 lack canonical signal peptides. However, in *P. falciparum*, PfiTPT is processed and cleaved, while PfoTPT is unprocessed due to its absence of a signal peptide (*Mullin et al., 2006*). The processing and topology of AMTs is complex and unpredictable in *T. gondii*, thus requiring further work to clarify the origin of either the processing hypothesis, alternative splicing, or alternative initiator codons in the parasite. Further analysis by IFA showed that the AID fusions were correctly localized to the apicoplast and were depleted within 24 hr of IAA treatment (*Figure 3B* and *Figure 3—figure supplement 2*); however, the parasite bodies were morphologically normal. To assess lytic growth of the AID lines, we assayed plaque formation of the AID parasites on host cell monolayers for 7 days. We observed that the AID lines grown in IAA did not form discernible plaques (*Figure 3C*), which was supported by further analysis of the plaque numbers and areas (*Figure 3D, E*). In summary, the AMTs are potentially essential for in vitro parasite growth.

We wondered if parasite replication contributed to the plaque defects. The parasites were grown in IAA for 24 hr, followed by scraping, purification and re-infection for the assay in the 2nd 24 hr and 3rd 24 hr. Using IFA analysis of the parasites, we observed that parasite replication was significantly affected in the first 24 hr of growth and became worse in the following cycles of 24 hr, especially the 3rd 24-hr period (*Figure 3F*). These results suggested that depletion of the transporters caused strong growth defects and delayed death of the parasites. We wondered if depletion of the proteins could cause loss or breakdown of the apicoplast, as reported previously for many apicoplast proteins (*Kennedy et al., 2019b*). ACP, an apicoplast marker, was used to test the apicoplast status (*Waller et al., 1998*). We observed that ACP partially diffused in the cytosol of intracellular parasites grown in IAA for 24 hr (*Figure 3—figure supplement 3A*), whereas it completely diffused in the cytosol of extracellular parasites that had grown in IAA for 24 hr (*Figure 3—figure supplement 3B*). This observation suggested swelling or breakdown of the apicoplast in the intracellular or extracellular parasites depleted with the AMTs, respectively. Detailed analysis of the phenotypes showed that the ACP diffusion gradually increased while comparing the three cycles for the same AID line in both the intracellular and extracellular parasites (*Figure 3G, H*). However, it was as low as ~30% in AMT1-AID, compared to those (60–70%) in AMT2-AID and dKD (*Figure 3G, H*). These results suggested that depletion of AMT2-AID resulted in much more severe defects (swelling or breakdown) of the apicoplast than that of AMT1-AID, and further depletion of AMT1-AID had no further effect.

## Depletion of the proteins resulted in defects in metabolic pathways

Based on the evolutionary predictions and features of MCTs, we suspected that the AMTs might be involved in transport of substrates, such as pyruvate, lactate, amino acids, as reported for human MCTs (*Felmlee et al., 2020*). As such, depletion of AMT1 and AMT2 would result in reduced levels of the metabolites isopentenyl pyrophosphate (IPP) and fatty acids in *T. gondii*. We first designed experiments to examine the possible defect of IPP synthesis in the AID lines upon induction, by assaying the capabilities of endocytic trafficking of green fluorescent protein (GFP) vesicles to the vacuolar compartment (VAC) (*Dou et al., 2014*). Our recent data showed that GFP transport to the

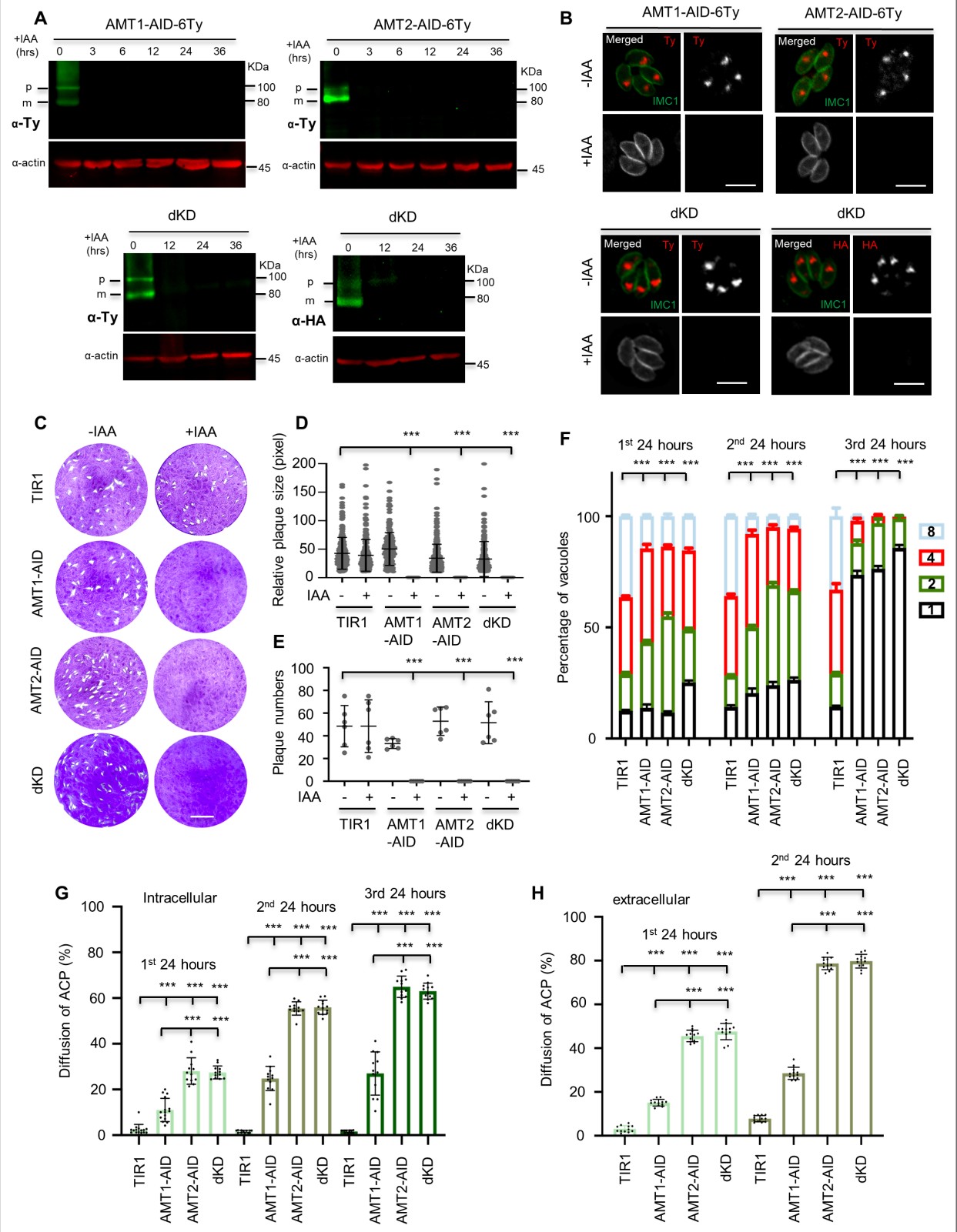

**Figure 3.** AMT1 and AMT2 are localized at the apicoplast membrane and essential for parasite growth in vitro. (**A**) Western blot detection of protein depletion in the IAA-inducible degron (AID) lines. The lines were induced by auxin (IAA) for different hours (hrs) as indicated. Western blots detected immatured (p) and matured (m) forms of the protein fusions in the non-induced lanes. Actin served as the control. (**B**) Indirect fluorescence assay (IFA) detection of protein depletion in the AID lines. Parasites were grown in IAA for 24 hr, followed by IFA with antibodies against Ty or HA (red) and IMC1

*Figure 3 continued on next page*

*Figure 3 continued*

(green). Scale = 5 µm. (**C–E**) Plaque formation by the TIR1 and AID lines on HFF monolayers in ±IAA for 7 days. Numbers (**D**) and sizes (**E**) of the plaques were measured by ImageJ. Scale = 0.5 cm. Two independent experiments with triplicates were performed. Data are shown as a mean ± standard error of the mean (SEM). (**F**) Parasite replication of the TIR1 and AID lines grown in IAA. The parasites were grown in IAA for 24 hr, followed by scraping, harvesting and infection for the 2nd and 3rd rounds of parasite growth in IAA. The parasites stained with GFP45 were counted in vacuoles (at least 150 vacuoles in each replicate). In comparing to TIR1 (1st, 2nd, and 3rd), $p < 0.0001$ for the AID lines with 2 and 8 parasites/vacuole in the 1st round, and for the AID lines with 1–8 parasites/vacuole in the 2nd round, $p < 0.0001$ for the AID lines with 1 and 4 parasites/vacuoles in the 3rd round. (**G, H**) Acyl carrier protein (ACP) diffusion in the TIR1 and AID lines grown in IAA. Parasites were grown in IAA for the 1st, 2nd, and 3rd lytic cycles, as described in the parasite replication assay (intracellular parasites). Those parasites in the 1st and 2nd round of growth were forced to egress for the same analysis (extracellular parasites). Vacuoles or single parasites were scored ($n > 150$ for each replicate). Fields/images were selected blind and all parasites/vacuoles were scored on the same fields/images (**F–H**). Three independent experiments with triplicates were performed (**A**, **B**, **C**, **F**, **G**, and **H**), and representative images were shown for **A**, **B**, and **C**, data are shown as a mean ± SEM with two-way analysis of variance (ANOVA) with Tukey's multiple comparisons (compared with the TIR1). \*\*\*$p < 0.0001$.

The online version of this article includes the following source data and figure supplement(s) for figure 3:

**Source data 1.** PDF file containing the uncropped western blot gels for verification of the IAA-inducible degron (AID) fusion lines in *Figure 3A*.

**Source data 2.** Excel file containing RAW images of plaque assays and scoring results of the parasite growth and apicoplast stability in *Figure 3*.

**Figure supplement 1.** AMT1 and AMT2 are co-localized in the apicoplast.

**Figure supplement 2.** Diagnostic PCR of the AMT1-AID, AMT2-AID, and dKD lines.

**Figure supplement 2—source data 1.** PDF file containing DNA gels for diagnostic PCR of the IAA-inducible degron (AID) fusion lines.

**Figure supplement 3.** Depletion of AMT1 and AMT2 leads to apicoplast defects and loss of mitochondrial membrane potential in parasites.

VAC following its uptake from host cell cytosol is dependent on protein prenylation of Rab1B and YKT6.1 (*Wang et al., 2023*). These proteins are modified at the C-terminal motif by isoprenoids, the synthesis of which could be specifically inhibited by atorvastatin at the host cells and by zoledronate in the parasite (*Li et al., 2013*), as demonstrated in a schematic and in positive experiments (TIR1 with two drugs at IC50 concentrations) with reduced GFP transport (*Figure 4A, B*). Therefore, measuring the endocytic trafficking of GFP vesicles in parasites in the absence/presence of atorvastatin could reflect levels of protein prenylation and isoprenoids (IPP). The substrates modified at the proteins were mainly derived from the apicoplast biosynthesis pathway – the MEP pathway. We then assayed the TIR1 and AID lines by growing them in GFP expressing HFF monolayers, where experimental groups were controlled by absence/presence of atorvastatin and IAA (*Figure 4B*), and where morpholinurea-leucine-homophenylalanine-vinylsulfone-phenyl (LHVS) was added to avoid degradation of GFP in the VAC. Following endocytosis of GFP vesicles by the micropore (manuscript in press), GFP vesicles were successfully transported to and accumulated in the VAC of TIR without IAA, with a high percentage (>20%) of parasites containing GFP foci (*Figure 4B*). In contrast, the GFP vesicles diffused through the cytosol, and were unable to accumulate at the VAC in the AID parasites grown in IAA with the absence/presence of atorvastatin, as shown by significantly reduced percentages (~8%) of the parasites in the AID lines induced for 12 hr (*Figure 4B*). We further tested Rab1B stability in the AMT-AID lines grown in IAA for 12 hr, which showed a strong defect of reduced protein abundance of Rab1B by scoring of parasites with normal or abnormal level of Rab1B using an IFA approach (*Figure 4—figure supplement 1*). This phenotype was observed in parasites with a reduced supply of isoprenoids by drug inhibition or knockdown of prenyl-transferase GGT-2 (*Wang et al., 2023*). These results altogether thus suggested that the IPP synthesis is significantly decreased in the AID lines depleted with the transporters. However, it was noteworthy that parasites depleted with AMT1-AID and AMT2-AID had similar levels of reduced GFP transport and parasites with reduced protein abundance of Rab1B at the 12 hr of IAA induction, suggesting that depletions of AMT1 and AMT2 resulted in similar levels of IPP reduction in parasites.

Our results in *T. gondii* reported a physiological effect that was observed in *P. falciparum*, where inhibition of the MEP pathway by fosmidomycin caused disruption of hemoglobin transport (*Howe et al., 2013*; *Kennedy et al., 2019a*). Herein, our results underscored the importance of the transporters for supporting the MEP pathway in the apicoplast. This was further manifested by a reduced level of mitochondrial membrane potential detected by mitotracker red in the AMT1-AID or AMT2-AID lines grown in IAA for 24 hr (*Figure 4—figure supplement 1*). IPP is the precursor for synthesis of ubiquinone, an essential electron carrier required for electron transport in the mitochondria (*Kennedy et al., 2019b*; *Sleda et al., 2022*). The reduced mitochondrial potential was also observed in the parasites

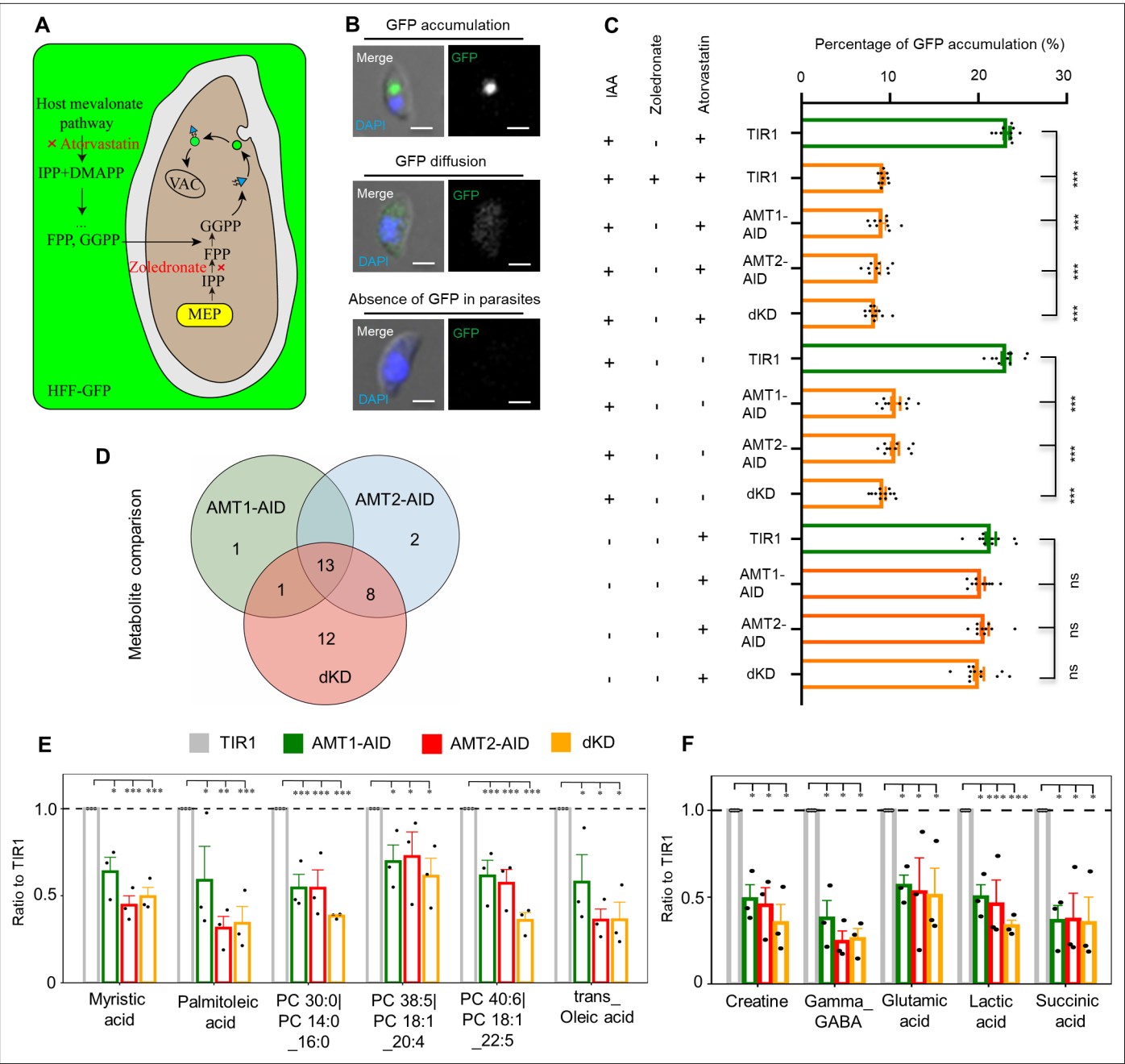

**Figure 4.** Depletion of AMT1 and AMT2 caused defects of methylerythritol phosphate (MEP) and fatty acid synthesis type II (FASII) pathways. (**A–C**) The MEP activity was strongly affected in the IAA-inducible degron (AID) lines. GFP transport to the VAC was assayed by growning parasties in GFP expressing HFF cells (HFF-GFP). LHVS (10 μM) was added to inhibit GFP digestion in the VAC (**Dou et al., 2014**). The isoprenoid synthesis in host cells and parasites was inhibited by IC50 concentrations of atorvastatin (10 μM) and zoledronate (0.3 μM) (**Li et al., 2013**), respectively (**A**). The inhibition over the TIR1 line using two of the drugs was used as a positive control, and representative images were shown for parasites with or withour GFP signal (**B**). The AID parasites were grown in HFF-GFP cells in the presence of LHVS for 36 hours in total, with the addition of atorvastatin, zoledronate or IAA as stated for different experiments in the figure in the last 12 hours. Parasites were immediately harvested, adhered and scored, as shown in the example images in B. Percentages of parasites with GFP foci were plotted (**C**). Fields/images were selected blind and all parasites/vacuoles were scored on the same fields/images (**C**). Three independent experiments with triplicates were performed. Data are shown as a mean ± SEM with one-way ANOVA with Tukey's multiple comparison. ***, $p < 0.0001$. Scale = 5 μm. (**D-F**) Fatty acid synthesis and TCA metabolites were significantly reduced in IAA induced AID parasites. Parasites were grown in IAA for 18 hours and harvested for untargeted metabolomics with GCMS and LCMS (N=3), and metabolites detected were analyzed by Dunnette's multiple comparison by comparing the AID parasites to the TIR1. Three AID lines shared 13 differential metabolites (**D**). These shared metabolites included fatty acids and lipids, such as myristic acid, palmitoleic acid, oleic acid, and phosphatidylcholine (PC) (**E**), and metabolites related to the TCA cycle (**F**). *, $p < 0.05$; **, $p < 0.001$; ***, $p < 0.0001$.

*Figure 4 continued on next page*

*Figure 4 continued*

The online version of this article includes the following source data and figure supplement(s) for figure 4:

**Source data 1.** Original images and scoring results of the GFP transport assay.

**Figure supplement 1.** Indirect effect of phenotypes resulted from depletion of AMT1 and AMT2.

depleted with APT1 (*Brooks et al., 2010*). Thus, the reduced mitochondrial membrane potential provided another piece of evidence supporting the notion of defective IPP synthesis in parasites depleted with the transporters.

We further examined activity of the FASII pathway in the TIR1 and AID lines grown in IAA for 18 hr, by testing metabolites (e.g. fatty acids [FAs]) in the parasites using untargeted metabolomics (gas chromatography–mass spectrometry [GC–MS] and liquid chromatography–mass spectrometry [LC–MS]). The metabolomics detected 66 metabolites in total in these lines (*Supplementary file 4a*). After analyses using Dunnett's multiple comparison methods, 15, 23, and 34 differential metabolites were identified in the parasite groups depleted with either AMT1-AID, AMT2-AID, or both (dKD) (*Supplementary file 4b*), in comparison with TIR1. Interestingly, three AID lines shared 13 differential metabolites (*Figure 4C*). The shared metabolites included fatty acids and lipids, such as myristic acid (C14 fatty acid), palmitoleic acid (C16 fatty acid), trans-oleic acid (C18 fatty acid), and phospholipids containing C14 and C18 fatty acids (*Figure 4D, E*). These results were consistent with previous studies where inactivated FASII proteins led to reduced synthesis of fatty acids up to 18 carbons long, and exogenous additions of the C14 and C16 fatty acids could complement growth defects (*Liang et al., 2020*; *Ramakrishnan et al., 2012*). Studies suggested that lipids can be partially acquired from the host cells (*Krishnan et al., 2020*), which includes those in the endocytic vesicles (*Koreny et al., 2023*; *Wan et al., 2023*). Here, we observed that the endocytic vesicles containing GFP were diffused in the parasite cytosol, indicating that the host cell-derived lipids were not affected in the protein-depleted parasites. We then concluded that depletion of the transporters significantly impaired the de novo biosynthesis of fatty acids by the FASII activity in the parasite. In contrast to the same level of reduced endocytic trafficking of GFP vesicles in the AID lines, an overall trend toward lower levels of myristic acid (C14 fatty acid), palmitoleic acid (C16 fatty acid), and trans-oleic acid (C18 fatty acid) was observed in AMT2-AID and tKD parasites in comparison with the AMT1-AID parasites (*Figure 4E*). These results supported the notion of a stronger impairment of the FASII pathway in parasites depleted with AMT2-AID.

Intriguingly, the metabolomics also identified metabolites related to the TCA cycle, such as gamma-aminobutyric acid (GABA), glutamic acid, and succinic acid (*Figure 4E*), which were shared among three of the AID lines. This would likely result from defective membrane potential caused by a reduced supply of IPP for synthesis of ubiquinone, as described above, or synthesis of heme that requires co-operation between the mitochondrion and the apicoplast (*Harding et al., 2020*). Another two metabolites, creatine and lactic acid, were likely associated with reduced metabolic levels in the parasites (*Figure 4E*). The untargeted metabolomics tends to identify differential metabolites by comparing the AID to TIR1 lines. Pyruvate was not identified, as it is actually a product of glycolysis in the cytosol. Collectively, the transporters are expectedly associated with the MEP and FASII pathways, to support synthesis of IPP and FA in the apicoplast.

## Depletion of AMT2 affects stability of ACC1

Depletion of AMT2 resulted in stronger defects on ACP diffusion and synthesis of fatty acids, suggesting a more important role of AMT2 in the apicoplast. To find out the reason for this, we first examined the protein level of one AMT (fused with 6 HA) in another AMT's AID-Ty line, thus assuring no reciprocal effect on protein level. After IAA induction for 36 hr, we did not see obvious changes to protein levels in the 6HA fusions in the AID-Ty lines on western blots and IFA (*Figure 5A, B*). Considering an MCT could transport biotin in mammalian cells (*Daberkow et al., 2003*), we wondered if AMT2 was able to transport biotin for activation of ACC1, which is a key enzyme in FASII (*Zuther et al., 1999*). ACC1 was endogenously fused with 6HA in the AID lines by a CRISPR approach. Upon protein depletion in IAA for 36 hr, ACC1 was found to be gone only in the AMT2-AID parasites, but not in the AMT1-AID parasites, as demonstrated by western blots (*Figure 5C*). Further analysis showed that ACC1 was gone in some parasites within a single vacuole in the AMT2-AID-depleted

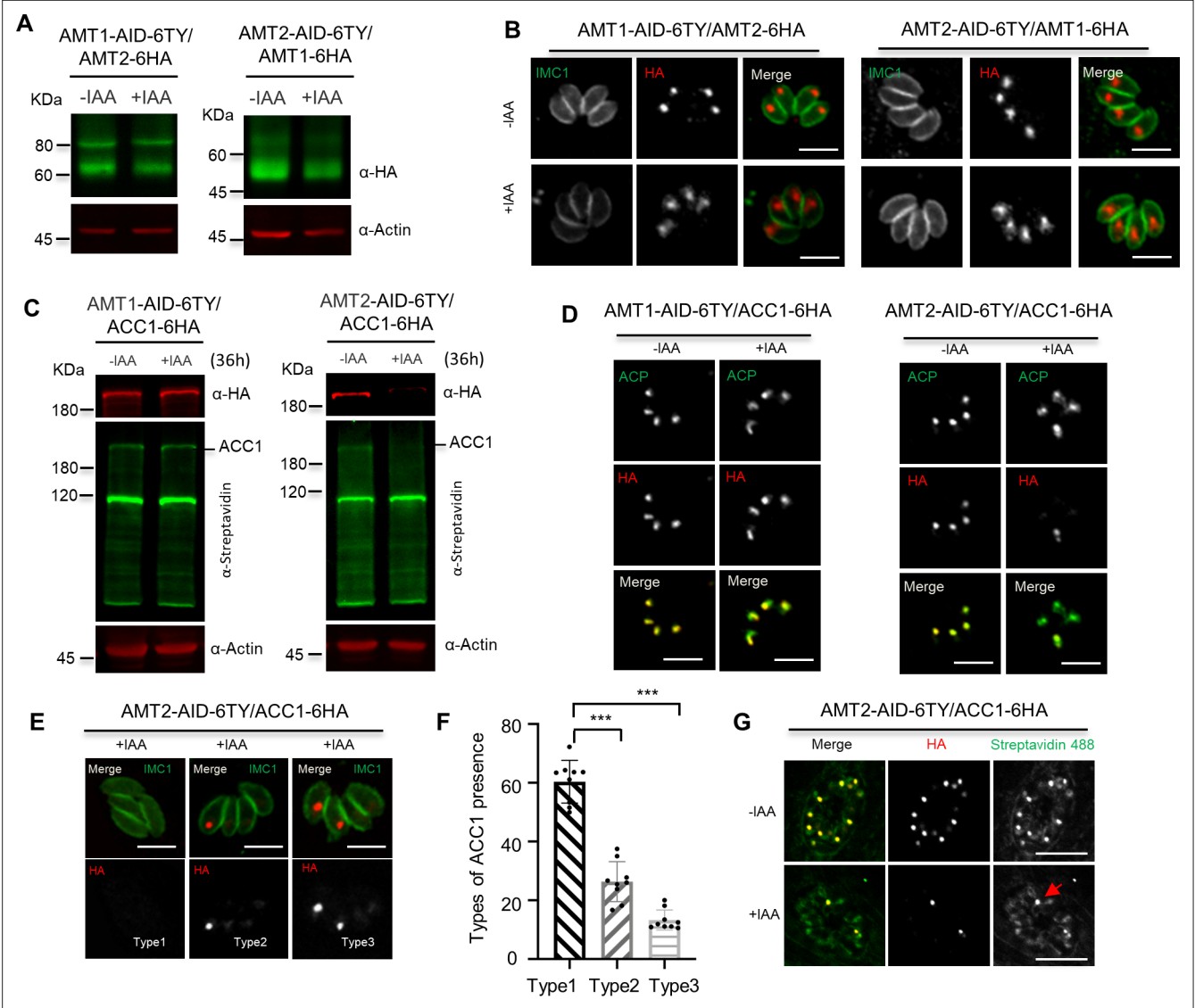

**Figure 5.** Depletiozn of AMT2 resulted in acetyl-CoA carboxylase 1 (ACC1) instability. (**A, B**) No reciprocal effect on protein levels by depletion of AMT1 or AMT2. AMT1 and AMT2 were reciprocally tagged at C-terminus with 6HA in the IAA-inducible degron (AID) lines, and the HA fusions were detected by western blots in parasites in ±IAA for 36 hr (**A**), and by indirect fluorescence assay (IFA) in parasites in ±IAA for 24 hr (**B**). IFA was performed with IMC1 (green) and Ty/HA (red). (**C, D**) ACC1 was lost upon depletion of AMT2-AID. ACC1 was endogenously tagged with 6HA at its C-terminus in the AMT1-AID and AMT2-AID lines, and parasites were grown in IAA for 36 hr for western blots (**C**) and IFA (**D**). IFA analyses were performed with anti-ACP and anti-HA antibodies, while western blots were done with anti-HA and streptavidin Li-COR 800CW. Actin served as the control. (**E, F**) Representative images of ACC1 on IFA in parasites of AMT2-AID induced in IAA for 24 hr. ACC1 (red) was completely lost in parasites of a vacuole (type 1), remained only in one parasite of a single vacuole (type 2), or in two parasites of a single vacuole (type 3) (**E**). The ACC1 types were scored in AMT2-AID induced in IAA for 24 hr, and expressed as percentages of different types (**F**). At least 150 parasites were scored in each replicate. Three experiments with triplicates were performed, and data are shown as a mean ± standard error of the mean (SEM) and analyzed by two-way analysis of variance (ANOVA) with multiple comparison, ***p < 0.0001. (**G**) The remaining ACC1 was still biotinylated. Parasites were grown in ±IAA for 24 hr for IFA analysis with antibodies against HA and streptavidin Alexa Fluor 488. Parasites containing both ACC1-6HA (red) and biotinylated ACC1 (green) was observed. Three independent experiments were performed with similar outcomes, and representative images were shown. Scale = 5 μm.

The online version of this article includes the following source data for figure 5:

**Source data 1.** PDF file containing the uncropped western blot gels of for strains in **Figure 5**.

parasites, where ACP was clearly stained with a slight diffusion (**Figure 5D**). Based on the presence or absence of ACC1 in parasites of the vacuoles, we can classify the vacuoles into three types – all parasites without ACC1 (type 1), one parasite (type 2), or two parasites (type 3) with ACC1 (**Figure 5E**). The quantification assay showed that only ~40% of the vacuoles contained one or two parasites with

ACC1 signal (types 2 and 3) (*Figure 5F*). However, ACC1 in those parasites was still biotinylated, as shown in streptavidin stains (*Figure 5G*). Collectively, AMT2 is associated with stability of ACC1 in a way, as yet, unknown, but unrelated with biotin availability in the apicoplast.

## AMT2 is essential to parasite virulence in mice

We next examined the in vivo growth of the AID parasites in mice administered with IAA. Prior to the mice assay, we attempted to gain further information about the ultra structures of the apicoplast upon depletion of the AMTs. The transmission electron microscopy (TEM) demonstrated that the four membranes surrounding the apicoplast remained visible, but were loosened, and the interior of the organelle appeared less dense and even electron lucent in the AID parasites, which had been induced for 24 hr in comparison with the TIR1 parasites (*Figure 6A*). We further quantified the apicoplasts with a reduced electron density in the AID parasites induced for 12, 24, and 36 hr vs the TIR1 in IAA for 36 hr, by examining 300 parasites in each time point. This examination showed that the apicoplast numbers identified by TEM dropped in the AMT1-AID parasites, when induced for 24 and 36 hr, and the numbers dropped by much greater levels in the AMT2-AID parasites, following 12 hr of IAA induction (*Figure 6B*). Notably, the apicoplast was identified only 3 times in the AMT2-AID parasites at 36 hr induction. Accordingly, the percentage of the abnormal apicoplast increased for both of the AID lines. In contrast, depletion of AMT2-AID resulted in much higher levels of abnormal apicoplasts at the three induction time points, compared to those in parasites depleted with AMT1-AID (*Figure 6B*). These results suggested that depletion of AMT1 caused modest loss of the apicoplast, yet the AMT2 depletion appeared to have a greater effect on the apicoplast loss.

The parental line TIR1 and its derivative lines (AMT1-AID, AMT2-AID, and dKD) were then used to intraperitoneally infect mice, at 100 parasites per mouse, followed by administering the mice orally and intraperitoneally with IAA in a solution on a daily basis. Those mice infected with TIR1 succumbed to lethal toxoplasmosis by day 10 post infection (*Figure 6C*). Conversely, the IAA treatment ultimately prevented acute virulence caused by the parasite lines with the AMT2 depletion (i.e. AMT2-AID and dKD), and it postponed mouse death by the AMT1-AID depletion. We next tested sera extracted from the surviving mice (which were infected by AMT2-AID) by performing IFA on *T. gondii* parasites. The IFA showed that the mouse sera clearly recognized proteins of the parasitophorous vacuoles of the parasites (*Figure 6—figure supplement 1*). We also further analyzed the efficacy of protein degradation of the AMT1-AID fusions in mice, by treatment with or without IAA for 24 hr after 5 days of growth, as illustrated in *Figure 6—figure supplement 1*. It was clear that AMT1-AID was efficiently degraded in mice by the treatment of IAA in only 24 hr (*Figure 6—figure supplement 1*). These results demonstrated that the mice infection assays were robust. However, the strong growth impairment conferred by AMT1 depletion in parasites in cell culture surprisingly resulted in only reduced parasite virulence in mice. We assume that this discrepancy is related to the relatively modest defects, such as the ACP diffusion/apicoplast abnormality (revealed by TEM)/FASII activity, as observed in the AMT1-depleted parasites. Thus, the remaining apicoplast is still functionally generating an adequate amount of metabolites, allowing the parasites to survive in mice. However, the discrepancy is possibly associated with different transport capabilities of substrates by AMT1 and AMT2. Though this does not perfectly explain the discrepancy, weaker phenotypes in AMT1-depleted parasites argue for this possibility. Collectively, AMT2 was obviously essential for establishing *T. gondii* infection and virulence in mice.

## Discussion

Most of the apicomplexan parasites, such as *T. gondii* and *Plasmodium* spp., harbor a vestigial plastid, which undergoes several fitness-conferring or essential metabolic pathways in the matrix (*Kloehn et al., 2021*; *Lim and McFadden, 2010*). However, the transporters that are responsible for exchanging metabolites between the apicoplast and the parasite remain elusive, due to difficulties in identification of those transporters in the multiple membranes that surround the organelle. Here, we fully exploited the currently available data, and the high-efficiency of proximity biotin labeling and CRISPR-Cas9 tagging approaches in the model apicomplexan *T. gondii*, and discovered several novel transporters in the apicoplast. Among those putative transporters, most of them appear to be derived from the endosymbiont (i.e. a red alga) in the evolution of the secondary plastid, and only

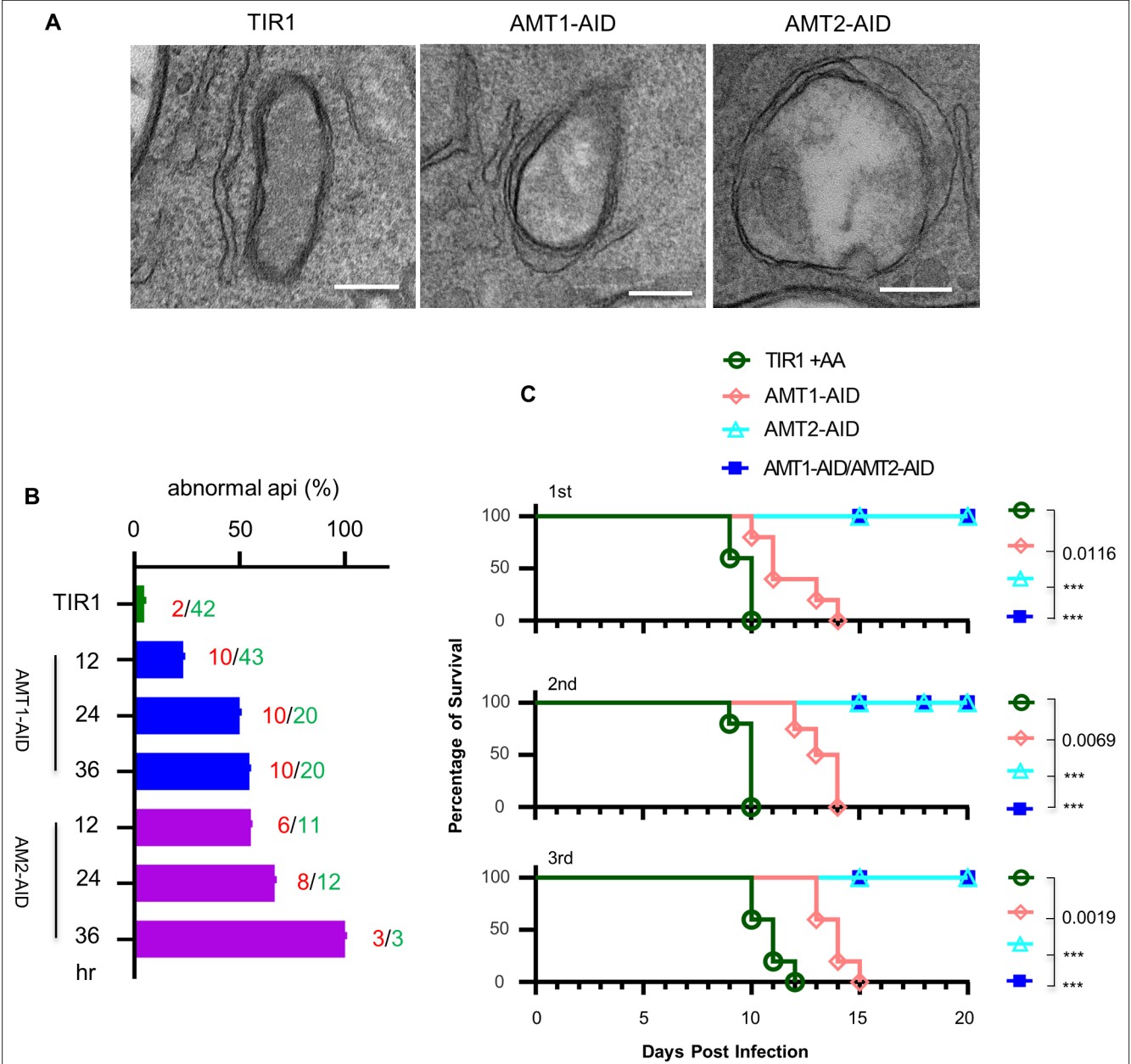

**Figure 6.** Examination of the apicoplast ultra structures, and mouse virulence of the parasites upon depletion of the IAA-inducible degron (AID) fusions. (**A**) Transmission electron microscopy (TEM) observed less electron dense or electron lucent lumen and loosened membranes in the apicoplast in the AID parasites induced in IAA for 24 hr. Representative images were shown. Scale = 200 nm. (**B**) Quantification of the apicoplast with the lumen less dense or lucent in parasites induced for 12, 24, and 36 hr and TIR in IAA for 36 hr. The TEM examined 300 parasites for three lines (TIR1, AMT1-AID, and AMT2-AID) at each time point, and numbers of abnormal apicoplast (red numbers) and normal apicoplast identified (green numbers) were shown by the columns. The columns express the percentages of abnormal apicoplast in all the identified apicoplasts. (**C**) Survival curve of BALB/C mice infected with 100 parasites intraperitoneally and treated with IAA for 20 days. The Gehan–Breslow–Wilcoxon test was used to compare differences between the survival curves (the AID line versus TIR1), ***p < 0.0001. The experiment of mouse infection was performed with five mice in each group, and results from three independent experiments (1st, 2nd, and 3rd) were shown together with symbols for the mouse groups.

The online version of this article includes the following source data and figure supplement(s) for figure 6:

**Source data 1.** Excel file containing the record of mouse infection assay using the IAA-inducible degron (AID) fusion lines.

**Figure supplement 1.** Confirmation of the mice assay by testing mice sera and protein degradation.

the pair of MCTs AMT1 and AMT2 is likely to be derived from the putative host cell that originally engulfed the endosymbiont. Protein depletion of the AMT1 and AMT2 using a highly efficient technique, the TIR1-AID system, resulted in defects of plaque formation in vitro, ACP diffusion/apicoplast loss and biosynthetic pathways of the MEP and FASII. However, depletion of AMT2 clearly caused stronger defects including ACP diffusion/apicoplast loss, FASII produced fatty acids, ACC1 stability and removal of mouse virulence. These results indicate that complex transporting functions are likely to exist in the apicoplast involving AMT1 and AMT2.

Recent studies have highlighted the difficulty in identifying apicoplast transporters, because of strong backgrounds in the proteomics resulting from high activity and nonspecific labeling of the biotin ligase (BioID) fused with apicoplast proteins (*Boucher et al., 2018*; *Sayers et al., 2018*). This remained true for our proximity proteomics from APT1-TurboID. Fortunately, our focused screening leveraged high efficiency by genomic editing in *T. gondii*, yielding positive results for 12 putative transporters. These novel transporters provided an important resource for evolution and functional analyses. Their conservation, including in plastid-deficient taxa, that is Gregarina and Cryptosporidia, indicates that these proteins were relics of the former plastid in evolution via endosymbiont gene transfers (*Gould et al., 2008*). Studies suggested that the metabolic integration of the primary plastid (e.g. chloroplast) was mainly driven by targeting transporters from a heterotrophic protist – the host cell, but not from the endosymbiont – cyanobacteria (*Karkar et al., 2015*; *Tyra et al., 2007*). However, our preliminary study in evolution indicated that most novel transporters were likely to be derived from the endosymbiont. In sharp contrast, the pair of MCTs (AMT1 and AMT2) may have an origin in a heterotrophic protist – the putative host cell. This observation therefore provides a perspective that the pair of AMTs may play critical roles in the metabolic integration of the algal endosymbiont into the parasites.

Our immuno-EM and confocal imaging provided a strong support for the membrane localization of the pair of transporters AMT1 and AMT2 in the apicoplast, indicating that the AMTs are likely to transport substrates in the membranes. We observed that the AMTs appeared to have two bands on western blot, which might be processed versions, or produced by alternative splicing or produced by an alternative initiator codon. Concerning the processing and targeting, apicoplast membrane proteins in *T. gondii* appear to be more complex than those in *P. falciparum*, by comparison to PfiTPT and PfoTPT (*Mullin et al., 2006*). Our phenotypic analysis showed that depletion of the transporters AMT1 and AMT2 caused strong defects on plaque formation and parasite replication, suggesting that both transporters are critical on the apicoplast membranes. However, we observed disparate degrees of phenotypic disruption in the AMT1- and AMT2-depleted parasites. Depletion of AMT2 always caused stronger defects in ACP diffusion, FASII fatty acids, apicoplast loss, and mouse virulence. AMT2 is additionally associated with loss of ACC1, a key enzyme for FASII, found in the majority of parasites in a vacuole, likely providing another piece of evidence to explain the greater phenotypic defects. Several attempts to delete ACC1 failed in our study, supporting a critical role of ACC1 in the FASII process. Genetic studies suggest that depletion of FASII enzymes led to loss of apicoplast (*Liang et al., 2020*). Thus, the observation of ACC1 loss supports stronger phenotypes in AMT2-AID-depleted parasites. In addition, depletion of the AMTs caused an initially gradual defect, but then a much more focused sharper defect in parasite replication in the 3rd 24 hr and showed a delayed death phenotype. Previous studies showed that parasite replication exhibited gradual deceases upon depletion of TPC and APT1 by utilization of a low efficient knockdown strategy – the tetracyclin-inducible system (TATi) (*Brooks et al., 2010*; *Li et al., 2021*). It is unknown whether the TATi system would mask the growth phenotype, yet, the AMT depletion appeared to have such a sharper change in the third cycle (24 hr in a growth cycle). Collectively, the AMTs are critical to parasites, yet AMT2 is more important to the apicoplast membrane function.

Based on knowledge of MCTs (*Felmlee et al., 2020*), the pair of AMTs is likely to transport pyruvate, lactate, amino acids, and other substrates. We thus suspect that AMT1 and AMT2 might be associated with transporting pyruvate and amino acids, since these are essential for the apicoplast metabolism. Pyruvate is the central metabolite that connects many metabolic pathways in living organisms. It must be imported into organelles for specific metabolic pathways, as suggested with pyruvate import by the NHD1/Bass2-protein coupling function in the chloroplast of C4 plants, and by the mitochondrial pyruvate carrier (*Bricker et al., 2012*; *Furumoto, 2016*; *Furumoto et al., 2011*; *Gasperotti et al., 2020*; *Herzig et al., 2012*). In *T. gondii*, and *P. falciparum*, studies have suggested

that the major pyruvate source has still not been established (*Niu et al., 2022*; *Swift et al., 2020*; *Xia et al., 2019*). Pyruvate is one of the key carbon sources for the condensation with triose phosphates to initiate the biosynthesis pathway of MEP in the parasite. Meanwhile, pyruvate is the only carbon source in the FASII pathway for biosynthesis of fatty acids (*Brooks et al., 2010*). The possible role on pyruvate import by the AMTs could provide a good explanation for the defects on MEP and FASII pathways, specifically, the phenotypes reflecting the reduced levels of IPP synthesis in the AMT-depleted parasites. We attempted to define this possible function of the AMTs by a pyruvate sensor PyronicSF in the parasite (*Arce-Molina et al., 2020*), and by a complementation of an *Escherichia coli* knockout strain that is deficient in pyruvate import (a generous gift from Prof. Kirsten Jung) (*Gasperotti et al., 2020*). However, several attempts with these approaches were not successful in generating the strain required for detailed examination of pyruvate import capability either in the parasite or in the *E. coli* strain.

Our work successfully identifies several novel apicoplast transporters, thus greatly unraveling the mystery of the apicoplast membrane composition and opening up possibilities to tackle difficult questions about metabolite transport in the apicomplexans. The most intriguingly findings in this study are the identification of the pair of MCTs AMT1 and AMT2, which are likely derived from the putative host cell that engulfed a red algal during the evolution of this group of parasites. Metabolic analysis together with other phenotypic analyses supported the notion that the pair of AMTs is associated with the MEP and FASII biosynthesis pathways. However, though both transporters belong to the MCTs, the phenotypic defects differ greatly, supporting a complex function for AMT2 in the apicoplast, however, the substrates transported by the AMTs still await further clarification. Collectively, the MCTs are likely to control similar essential processes in related protists, especially in other members of this phylum of deadly parasites.

## Limitations of this study

This study has begun to look into the possibly evolutional origins of newly identified transporters although, as yet, observations are preliminary and await further analysis to clarify specific questions, as demonstrated in the detailed evolutionary analysis of chloroplast transporters (*Karkar et al., 2015*). Our further analysis on the phenotypes of two of these transporters, AMT1 and AMT2, using parasites depleted with these activities, demonstrates differing degrees of defects at critical points in the parasite's physiology and metabolism. We also observed similar phenotypes reflecting decreased levels of the MEP biosynthesis activity upon depletion of AMT1 and AMT2, indicating that phenotypic outcomes did not result from apicoplast abnormality, since other phenotypic outcomes were more severe in the AMT2-depleted parasites. We predict, therefore, that the metabolic defects possibly result from a reduced supply of substrates by each of the transporter. It is noteworthy that other studies provided us with comparable knowledge in phenotypes in TPC- and APT1-depleted parasites. In these parasites, metabolic defects in the apicoplast pathways were observed by indirect measurements as well (*Brooks et al., 2010*; *Li et al., 2021*). However, these phenotypes were possibly caused by pleiotropic effects of apicoplast loss, since they were analyzed by a comparably slow inactivation system to deplete the transporters of TPC and APT1 (with an induction of at least 3–4 days). Nevertheless, in our case with a highly efficient inactivation system, we still cannot exclude the effect of apicoplast loss on phenotypic results caused by depletion of the AMTs. Therefore, a more precise understanding of the functions of the AMTs will require further identification of the topology, protein interactions and substrate transport capability in the complex configuration of the apicoplast membranes. Our work, as yet, unsuccessful, in attempting to use a pyruvate sensor and/ or a heterologous expression the AMTs in *E. coli*, is the first attempt to examine the transport capability of pyruvate by AMT1 and AMT2. Thereafter, the application of more sophisticated approaches, such as *Xenopus* oocytes, an improved *E. coli* system, or a yeast expression system, may enable us to tackle the important question of substrate transport by AMT1 and AMT2. Future progress in this direction will thus shed critical insights into the discrepancies in parasite growth that we observe when comparing cell culture versus infections in mice.

# Materials and methods

## Key resources table

| Reagent type (species) or resource | Designation | Source or reference | Identifiers | Additional information |
|---|---|---|---|---|
| Gene (*Toxoplasma gondii*) | As listed in **Supplementary file 2** | ToxoDB release 53 | **Supplementary file 2** | |
| Strain, strain background (*T. gondii*) | *T. gondii* RHΔ*hxgprt*Δ*ku80*; RHΔ*hxgprt*Δ*ku80*/TIR1 and **Supplementary file 1b** | Generous gifts from Prof. Vern Carruthers and Prof. David Sibley PMID:19218426 PMID:28465425 | | See Materials and methods |
| Strain, 6- to 8-week-old femal, Balb/C (*Mus musculus*) | Balb/C | Division of Experimental Animals, Beijing University | Strain code 028 | |
| Recombinant protein | **Supplementary file 2c** | This paper | **Supplementary file 2c** | See Materials and methods |
| Softwares, algorithm | PRISM, version 8.0 | http://www.graphpad.com | GraphPAD v8.0 RRID:SCR_002798 | |
| Softwares, algorithm | InterPro:protein sequence analysis & classification | https://www.ebi.ac.uk/interpro/ | InterPro RRID:SCR_006695 | |
| Softwares, algorithm | *Toxoplasma* Genomics Resource | http://toxodb.org/toxo/ (**Alvarez-Jarreta et al., 2024**) | ToxoDB release 53 | |
| Softwares, algorithm | Eukaryotic Pathogen Genomics Resources | https://veupathdb.org/veupathdb (**Alvarez-Jarreta et al., 2024**) | VEuPathDB | |
| Softwares, algorithm | MetaboAnalyst 5.0 | https://www.metaboanalyst.ca (**Pang et al., 2021**) PMID:34019663 | MetaboAnalyst | |
| Softwares, algorithm | jackhmmer | http://hmmer.org/ (**Prakash et al., 2017**) PMID:29220076 | HMMER v3.3.2 | |
| Softwares, algorithm | trimAL v1.2.rev57 | https://vicfero.github.io/trimal/ (**Capella-Gutiérrez et al., 2009**) PMID:19505945 | trimAL v1.2.rev57 | |
| Softwares, algorithm | TBtools | https://github.com/CJ-Chen/TBtools (**Chen and Zhao, 2023**) PMID:37740491 | | |
| Softwares, algorithm | R v4.2.0 | https://www.r-project.org/ | RRID:SCR_001905 | |
| Softwares, algorithm | ComplexHeatmap | https://bioconductor.org/packages/release/bioc/html/ComplexHeatmap.html (**Gu et al., 2016**) PMID:27207943 | | |
| Softwares, algorithm | ChiPlot | https://www.chiplot.online/ | | |
| Softwares, algorithm | Mafft v7.490 | https://mafft.cbrc.jp/alignment/software/ (**Katoh et al., 2019**) PMID:28968734 | | |
| Chemical compound, drug | Pyrimethamine | Sigma-Aldrich | Cat#46706 | 3 µM |
| Chemical compound, drug | Mycophenolic acid | Sigma-Aldrich | Cat#M5255 | 25 µg/ml |
| Chemical compound, drug | Xanthine | Sigma-Aldrich | Cat#X4002 | 25 µg/ml |
| Chemical compound, drug | 3-Indoleacetic acid (auxin) | Sigma-Aldrich | Cat#I2886 | 500 µM |

*Continued on next page*

*Continued*

| Reagent type (species) or resource | Designation | Source or reference | Identifiers | Additional information |
|---|---|---|---|---|
| Chemical compound, drug | D-biotin | Sigma-Aldrich | Cat#B4639 | 500 µM |
| Chemical compound, drug | 4% paraformaldehyde solution | Solarbio PMID:37108334 PMID:36813769 | Car#P1110 | |
| Antibody | Mouse anti-Ty (BB2) | In house hybridoma; gift from Prof. Philippe Bastin | | 1:100 |
| Antibody | Rabbit polyclonal anti-HA (SG77) | Thermo Fisher Scientific | Cat#71-5500 | 1:500 |
| Antibody | Mouse monoclonal anti-HA | BioLegend | Cat#901501 | 1:500 |
| Antibody | Rabbit polyclonal anti-GAP45 | Home-made PMID:36813769 | Polyclonal | 1:500 |
| Antibody | Rabbit polyclonal anti-actin | Home-made PMID:36813769 | Polyclonal | 1:1000 |
| Antibody | Rabbit polyclonal anti-IMC1 | Home-made PMID:36813769 | Polyclonal | 1:1000 |
| Antibody | Rabbit polyclonal anti-ACP | Home-made PMID:36813769 | Polyclonal | 1:1000 |
| Antibody | Mouse monoclonal anti-GFP | Thermo Fisher Scientific | Cat#MA5-15349 | 1:500 |
| Antibody | Streptavidin Alexa Fluor-488 conjugate | LICOR | Cat#926-3230 | 1:2000 |
| Antibody | Alexa Fluor 488 Goat anti-mouse IgG (H+L) | Thermo Fisher Scientific | Cat#A-11029 | 1:2000 |
| Antibody | Alexa Fluor 568 Goat anti-mouse IgG (H+L) | Thermo Fisher Scientific | Cat#A-11031 | 1:2000 |
| Antibody | Alexa Fluor 568 Goat anti-rabbit IgG (H+L) | Thermo Fisher Scientific | Cat#A-11036 | 1:2000 |
| Antibody | Alexa Fluor 488 Goat anti-rabbit IgG (H+L) | Thermo Fisher Scientific | Cat#A-11034 | 1:2000 |
| Antibody | IRDye 800CW Goat anti-mouse IgG (H+L) | LI-COR | Cat#926-32210 | 1:2000 |
| Antibody | IRDye 800CW Goat anti-rabbit IgG (H+L) | LI-COR | Cat#926-32211 | 1:2000 |
| Antibody | IRDye 680CW Goat anti-rabbit IgG (H+L) | LI-COR | Cat#926-68071 | 1:2000 |
| Antibody | IRDye 680CW Goat anti-mouse IgG (H+L) | LI-COR | Cat#926-68070 | 1:2000 |
| Antibody | IRDye 800CW Streptavidin | LI-COR | Cat#926-32230 | 1:2000 |
| Commercial assay or kit | pEASY-Basic Seamless Cloning and Assembly Kit | Transgene | CU201-02 | |
| Commercial assay or kit | Pierce Streptavidin Magnetic Beads | Thermo Fisher Scientific | Cat#88816 | |
| Commercial assay or kit | ProLong Gold Antifade Mountant without DAPI | Thermo Fisher Scientific | Cat#36930 | |
| Commercial assay or kit | ProLong Gold Antifade Mountant with DAPI | Thermo Fisher Scientific | Cat#36931 | |
| Cell line (include species here) | Human Foreskin Fibroblasts (human) | ATCC | Cat#SCRC-1041 | |
| Others | Fetal bovine serum (solution) | Transgene PMID:37108334 PMID:36813769 | FS201-02 | |

*Continued on next page*

*Continued*

| Reagent type (species) or resource | Designation | Source or reference | Identifiers | Additional information |
|---|---|---|---|---|
| Others | Dulbecco's modified Eagle medium (DMEM) (powder) | Thermo Fisher Scientific PMID:37108334 PMID:36813769 | Cat#12800-082 | |
| Others | Penicillin–streptomycin (100×) | Solarbio PMID:37108334 PMID:36813769 | P1400 | |
| Others | Glutamine (100×) | Solarbio PMID:37108334 PMID:36813769 | G0200 | |

## Parasite and cell culture

*T. gondii* RHΔ*hxgprt*Δ*ku80* (**Huynh and Carruthers, 2009**), RHΔ*hxgprt*Δ*ku80*/TIR1 (**Brown et al., 2017**) and their derivative lines are listed in **Supplementary file 1b–d**. The tachyzoite lines were grown in human foreskin fibroblast (HFF-1 from ATCC SCRC-1041) in D5 media prepared by DMEM supplemented with 5% inactivated fetal bovine serum (Transgene, FS201-02), 10 mM glutamine and penicillin–streptomycin. The lines and HFF-1 were maintained mycoplasma negative as described previously (**Long et al., 2016**). The TIR1 and AID lines were cultured in HFF-1 with 500 μM auxin (+IAA) or 0.1% ethanol alone (−IAA) for phenotypic assays, as previously described (**Brown et al., 2017**; **Long et al., 2017b**). Parasites were harvested by passing through 22 g needles and by filtration through 3.0 μm polycarbonate membranes, from which extracellular parasites were used for experimental assays.

## Mice

*T. gondii* parasite growth in vivo was tested in 6-week-old BALB/C mice. Mice were maintained under specific pathogen-free (SPF) conditions in filter-top cages with provision of sterile water and food. Mice were randomly assigned to experimental groups (*n* = 5 per group) and challenged by intraperitoneal (i.p.) injection of 100T. *gondii* tachyzoites, followed by monitoring on a daily basis to record their health by checking their appearance, body weight, and responsiveness. The mouse experiments were conducted according to the guidelines and regulations issued by the Veterinary Office of the China Agricultural University (Issue No. AW11402202-2-1).

## Generation of plasmids and *T. gondii* lines

A CRISPR/Cas9 sgRNA 3′ plasmid (**Supplementary file 1c**) targeting a region close to a stop codon at a specific gene can efficiently produce Cas9 and sgRNA to create DNA double strand breaks (DSB) in parasites, which facilitates the integration of a tagging amplicon at the 3′ terminus of a specific gene. The sgRNA-specific sequences were selected, as described in our previous protocol (**Brown et al., 2018a**). The sequences were integrated into the pCas9 plasmid using a basic seamless cloning and assembly kit (CU201-02, TransGen Biotech) with primers listed in **Supplementary file 1d**, as described in our recent study (**Wan et al., 2023**). The amplicon for a specific integration at the DSB was generated from a generic tagging plasmid (**Supplementary file 1c**) using a pair of primers L and T (**Supplementary file 1d**), as described in our previous study (**Long et al., 2017b**), which was illustrated in **Figure 4—figure supplement 1A**. These generic plasmids included pL-6Ty-HXGPRT (**Brown and Sibley, 2018b**), pL-6HA-DHFR (**Long et al., 2017a**), pL-6HA-HXGPRT (addgene #86552) (**Long et al., 2017b**), pL-TurboID-4Ty-DHFR (**Wan et al., 2023**), pL-AID-6Ty-DHFR (**Wan et al., 2023**), and pL-AID-3HA-HXGPRT (addgene #86553) (**Long et al., 2017b**). One additional generic plasmid pL-AID-GFP-HXGPRT was generated using pL-AID-3HA-HXGPRT as the PCR template of plasmid backbone, followed by cloning the GFP fragment into the backbone with primers listed in **Supplementary file 1d**. The amplicon generated by primers L and T from the generic plasmids contains short homologs (HR1 and HR2, 41 bp) for targeting the upstream (HR1) of the stop codon and the downstream (HR2) of the Cas9 cleavage site (DSB), resulting in integration of epitope tags or AID fragments in frame with the encoding sequences of specific genes. The drug-selected lines were

sub-cloned, followed with screening with diagnostic PCR, IFA and western blots. The resistance markers containing LoxP sites can be excised by transfection with pmini-Cre, as previously described (*Heaslip et al., 2010*).

## Transfection and selection

Parasites were harvested and resuspended in Cytomix (10 mM $KPO_4$, 120 mM KCl, 5 mM $MgCl_2$, 25 mM HEPES, 2 mM EDTA) $1 \times 10^8$ parasites/ml, then combined with 10–20 µg pCas9-sgRNA 3' and 1–5 µg amplicons with 2 mM ATP, 2 mM GSH, and 150 µM $CaCl_2$ to a final volume of 250 µl, followed by electroporation using a BTX ECM 830 electroporator (Harvard Apparatus), as described previously (*Soldati and Boothroyd, 1993*). Parasites were selected on the second day with addition of appropriate drugs mycophenolic acid (MPA) (25 µg/ml) and 6-xanthine (6Xa) (50 µg/ml), or pyrimethamine (Pyr) (3 µM). Negative selection was applied with 6-thio-xanthine (200 µg/ml) when parasites were transfected with a pCre plasmid to remove the resistant marker HXGPRT. Positive selection pools of parasites were subcloned in 96-well plates at 3 parasites/well and allowed to grow for 6–7 days before selection of wells containing a single plaque. Pure clones were diagnosed by PCR using appropriate primers (*Supplementary file 1d*) and IFA analysis using appropriate antibodies if applicable.

## Indirect fluorescence assay

Parasites were fixed by 4% paraformaldehyde in phosphate-buffered saline (PBS), followed by blocking and permeabilization using PBS containing 2.5% bovine serum albumin (BSA) and 0.25% Triton X-100. Antibody incubation was performed with indicated combinations of primary antibodies for 30 min, followed by incubation with secondary antibodies conjugated with fluorescent dyes (Thermo Fisher Scientific) for 30 min. The above procedures were applied to the ACP diffusion analysis for parasites as stated in the results and legends. The blind counting was conducted for at least 150 vacuoles (intracellular parasites) or single parasites (extracellular parasites) in each replicate. ProLong Gold Antifade Mountant with or without DAPI were used to mount the coverslips on glass slides. Parasites were then imaged using a Nikon Ni-E microscope C2+ equipped with a DS-Ri2 Microscope Camera. The co-localization analyses were performed using the Ni-E software system NIS Elements AR. All the IFA analyses were carefully repeated at least three times, and similar results were obtained for the same experiments before finalizing the analysis.

## Western blotting

Parasites were resuspended in PBS, followed by addition of 5× Laemmli sample buffer supplemented with 1 mM D-Dithiothreitol (DTT) for a further incubation at 98°C for 10 min. Protein lysis was then resolved in sodium dodecyl sulfate–polyacrylamide gel electrophoresis (SDS–PAGE), blotted by a Bio-Rad wet-blotting system, subsequently incubated with different combinations of primary and secondary antibodies, followed by secondary antibodies conjugated with LI-COR reagents, or streptavidin LI-COR 800CW. The blot membranes were then visualized using a Bio-Rad ChemiDOC MP system.

## Turboid and biotinylated proteins

The APT1-TurboID line was grown on HFF monolayers for 24 hr, subsequently incubated with 500 µM D-biotin for 90 min. The parasites were fixed with 4% paraformaldehyde solution for IFA or harvested for western blots. IFA and western blots were performed with appropriate streptavidin reagents to visualize biotinylated proteins in parasites. The parental line and APT1-TurboID line were grown in HFF cells for 36 hr, followed with labeling of proteins by 500 µM D-biotin for 90 min, and harvesting of the parasites for purification of biotinylated proteins. In parallel, the APT1-TurboID parasites after 36 hr growing were harvested, and extracellular parasites were incubated with 500 µM D-biotin for 90 min, followed with by collection of parasites for purification of biotinylated proteins. The parasites were lysed in a buffer containing 1% Triton X-100, 0.2% SDS, and 0.5% deoxycholate, and sonicated with a microtip in 550 sonic dismembrator (Thermo Fisher Scientific). Biotinylated proteins in cleared supernatant were purified using streptavidin magnetic beads, exactly following the protocol described elsewhere (*Long et al., 2018*).

## Proteomic analysis

The purified proteins were separated on SDS–PAGE and stained by Coomassie blue R250, 45% methanol and 10% glacial acetic acid, subsequently dried on SDS–PAGE slices by vacuum. The dried gels were rehydrated, alkylated, and washed to remove the stain and SDS, and subjected to steps of mass-spectrometry analysis, following the protocol described in the Bio-Protocol (*Long et al., 2018*). The parental line and TurboID fusion line were analyzed in parallel with three technical and biological replicates. Resulting spectra were searched against a combined database of *T. gondii* ME49 (http://ToxoDB.org, release 29) and human proteins and a decoy database, using Mascot and Scaffold. The current views in Scaffold were exported into an excel spreadsheet using the following settings of peptide 2, protein threshold 99% and peptide threshold 95%. Hits were analyzed using embedded tools in ToxoDB (https://toxodb.org/toxo/app/) to predict the TMD. Peptide numbers of the hits in the TurboID were compared to those in the parental line (0 peptide was considered as 1), and fold changes were used to plot a volcano map against $Log10^{p\ value}$ in ggplot2 package in R.

## Conservation and phylogenetic analysis

The apicoplast proteins identified in this study and known apicoplast proteins as listed in the results (*Supplementary file 3a*) were used as queries for searching orthologs in organisms in Eukaryote, including species belonging to the chromalveolates, plants, metazoan, and fungi, and Bacteria (*Supplementary file 3a*). These proteins were simultaneously searched with the profile HMMs search engine (jackhammer). The *E*-value cutoff was set at 1e−7 and hits with an *E*-value <1e−7 were considered as orthologs. The best hits from the representative species in above taxa were merged and resultant list of orthologs (*Supplementary file 3b*) were used for conservation analysis with ComplexHeatmap package in R (v4.2.0) (*Gu et al., 2016*). This analysis pulled up the protein conservation heatmap, with indication by a heat scaled with *E*-value for each individual ortholog. The tree dendrogram without meaningful branch lengths across these species, and across the proteins were clustered according to the Euclidean distance using UPGMA method. Representatives from the known apicoplast proteins were analyzed in parallel as the control for comparison.

## Plaque formation

Parasites (150 parasites) were inoculated on confluent HFF monolayers in one 6-well in plates, followed with addition of either 500 µM IAA or ethanol alone (0.1%) and growth in D5 media at 37°C for 7 days. The parasites and monolayers were then fixed with 70% ethanol for 15 min, stained with 0.5% crystal violet for 5 min, subsequently washed with dH$_2$O and dried at room temperature. The plaque monolayers were recorded by scanning using a HP Scanjet G4050, and plaque numbers and plaque sizes were measured in ImageJ.

## Parasite replication

The TIR1 and its derived AID lines were grown in 500 µM IAA for 24 hr on monolayers in 24-well plates with coverslips. The parasites were fixed with 4% paraformaldehyde for 10 min, followed with blocking and permeabilization with 0.25% Triton X-100 in PBS containing 2.5% BSA. The parasites were then incubated with GAP45 antibodies and secondary anti-rabbit antibodies conjugated with Alexa Fluor 568. The coverslips were mounted and visualized under a Nikon Ni-E microscope C+. Vacuoles containing different numbers of parasites were scored on the same images with blindness, and at least 150 vacuoles were counted in each replicate of three independent experiments. Ratios of vacuoles containing different numbers of parasites were plotted against the total vacuoles examined.

## GFP acquisition assay

HFF cells expressing cytosolic GFP were prepared in our previous study (*Wan et al., 2023*). In brief, pLVX-EF1a-GFP-IRES-puromycin was combined with packaging plasmids psPAX2 (addgene #12260) and pMD2.G (addgene #12259) in Opti-MEM I (31985062, Gibco), and mixed with Fugene HD (E2311, Promega), and transfected into 293T cells. Viruses were collected and mixed with polybrene in a 12-well plate well containing 30–50% confluent HFF-1 cell monolayers. After 24 hr of post-transduction, the cell monolayers were re-seeded with incubation of puromycin (2.5 µg/ml) for 3 days. The cell monolayers were then maintained and expanded in D5 media. The TIR1 and AID lines were grown in the cell monolayers with IAA, 10 µM LHVS and 12.5 µM atorvastatin for 12, 24, or 36 hr.

LHVS inhibits cysteine protease CPL to accumulate GFP derived from host cell cytosol, as reported previously (*Dou et al., 2014*), Atorvastatin can inhibit synthesis of farnesyl pyrophosphate (FPP) and geranylgeranyl pyrophosphate (GGPP) in host cells, to decrease the supply of FPP and GGPP from the host cells to the parasites. In parallel, TIR1 was grown in the host cells with 10 µM LHVS, 0.3 µM zoledronic acid, and 12.5 µM atorvastatin, serving as a positive control to observe defects in GFP transport by decreasing protein prenylation in parasites. The addition of atorvastatin and zoledronic acid could specifically inhibit the FPP and GGPP supply both in host cells and parasites (*Li et al., 2013*), thus inhibiting GFP transport to the VAC in parasites, as demonstrated in *Figure 5A, B*. Purified parasites were adhered onto poly-lysine-coated coverslips, followed with fixation with 4% paraformaldehyde for 10 min. The parasites were permeabilized by 0.25% Triton X-100 in PBS, subsequently mounted in ProLong Gold Antifade Mountant with DAPI for visualization under a Nikon Ni-E microscope C+. Parasites were judged by observations of parasites with either GFP foci, GFP diffusion, or GFP absence, as demonstrated in *Figure 4B*. Fields/images were selected blind, and all parasites in the same images were counted for scoring of parasites with or without GFP signal. The counting was performed in replicate for at least 150 parasites. Percentages of parasites containing GFP foci were calculated and expressed as percentages in the populations.

## Immuno-staining for electron microscopy

Parasites were grown on HFF monolayers for 24 hr, subsequently digested with trypsin for collecting samples. Samples were fixed in a 0.1 M sodium cacodylate buffer (pH 7.4) containing 3% (wt/vol) paraformaldehyde, 0.1% glutaraldehyde (vol/vol), and 4% sucrose (wt/vol) at 4°C overnight. The samples were then washed four times by a 0.1 M sodium cacodylate buffer containing 4% (wt/vol) sucrose and neutralized on ice by a 0.1 M sodium cacodylate buffer containing 4% (wt/vol) sucrose, 0.1 M glycine, and dehydrated in ethanol. The samples were immersed in LR White resin (Ted Pella, Inc) at 4°C overnight, subsequently polymerized at light intensity $1.2 \times 10^5$ µJ/cm$^2$ on ice for 5 hr. The LR White imbedded samples were sectioned with a Leica ultramicrotome (EM UC7) at 100 nm and were picked onto 100-mesh formvar-carbon-coated nickel grids. The loaded nickel grids were then blocked and incubated with mouse anti-Ty antibodies (1:5) at RT for 2 hr and overnight at 4 °C, followed with washing by PBS. The samples on grids were incubated with Alexa Fluor 488 goat anti-mouse IgG 10 nm colloidal gold particle (1:100) (15 or 10 nm in size) for 1 hr, followed with washes in PBS and double-distilled water. The samples on grids were stained by 3% phosphotungstic acid and observed under a JEM1400FLASH transmission electron microscope operation at 120 kV.

## Transmission electron microscopy

Parasites grown on HFF cell monolayers were collected by scraping off, fixed by a fixative mixture of 2% paraformaldehyde and 3% glutaraldehyde at 4°C overnight. The samples were washed, fixed in 1% osmium tetroxide in 50 mM phosphate buffer at 4°C for 1 hr, washed three times and dehydrated by sequential concentration of ethanol, and by acetone and Spurr for 1 hr. The samples were then imbedded in Spurr and sectioned to 100 nm slices by a Leica ultramicrotome (EM UC7). The ultrathin sections were collected on 100 mesh grids. The grids were then viewed by JEOL transmission electron microscopy (JEM1400FLASH).

## Untargeted metabolomics

Parasites were grown in IAA for 18 hr and purified in chilled PBS, followed by immediate quenching by addition of −40°C methanol and sterile ddH$_2$O (400 µl methanol + 100 µl H$_2$O). The samples were processed for the analysis of metabolites by GC–MS and LC–MS, exactly following the protocol reported in our recent study (*Wan et al., 2023*). Briefly, the GC–MS analysis was performed with An OPTIMA 5 MS Accent fused-silica capillary column on Agilent7890A/5975 C GC–MS system (Agilent Technologies Inc, CA, USA), while the LC–MS analysis was performed on a Thermo Fisher Ultimate 3000 UHPLC system with a Waters ACQUITY UPLC BEH C18 column (2.1 mm × 100 mm, 1.7 µm) using standard positive and negative modes. For GC–MS, the peak picking, alignment, deconvolution, and further processing of resulting raw data were carried out according to the previously published protocols (*Gao et al., 2010*). The final data were exported as a peak table file, including observations, variables (rt_mz), and peak areas. The data were normalized against the total peak value of total peaks, and differential metabolites were identified by statistical analysis performed in R platform, where

parametric and nonparametric tests were done using one-way analysis of variance (ANOVA) with Dunnett's multiple comparison, to identify differential metabolites with $p < 0.05$ and fold change $>1.2$ for further analysis. For LC–MS, the raw data were transformed to mzXML format by ProteoWizard and then processed by XCMS and CAMERA packages in R software platform. The final data were exported as a peak table file, including sample names, variables (rt_mz) and peak areas. The peak areas data were normalized to internal standards before the analysis of statistics. The in-house data-base was referenced to identify the compound by using $m/z$ (MS1), mass spectra (MS2), and retention times (RT). Normalization for relative parasite amounts was based on the total integrated peak area values of metabolites within an experimental batch. Metabolites identified by the GC–MS and LC–MS were merged together for further analysis (*Supplementary file 4a*).

## Parasite growth in mice

Mice were randomly assigned to groups and were intraperitoneally injected with 100 parasites per mouse. Subsequently, mice ($n$ = 5 mice for each line) were orally administered with IAA (3-indoleacetic acid, auxin, Sigma) and by an intraperitoneal injection on a daily basis (*Brown and Sibley, 2018b*; *Yesbolatova et al., 2020*). The drinking water contained sterile water with 1 mg/ml IAA, 3 mM NaOH, 5% sucrose (wt/vol), flavored with 2 mg/ml TANG (Mondelēz International), pH 8.0. Meanwhile, mice were intraperitoneally injected with a daily 0.2 ml of sterile solution containing 15 mg/ml IAA, 1 M NaOH, pH 7.8. The mice infected with the TIR1 line served as the control, which received the drinking water and injection of IAA-containing solution in the same way to the AID lines. The mice were weighed and their health conditions were monitored on a daily basis until day 20.

## Quantification and statistical analysis

Experiments were conducted blind for at least two to three biological replicates and statistical analyses were conducted in Graphpad v8.0. One- or two-way ANOVA with Tukey's multiple comparison was conducted for data that fit normal a distribution, while one-way ANOVA with Dunnett's multiple comparison was tested for data with small sizes or abnormal distributions. $p < 0.05$ was considered significant. Experiment-specific statistical information is provided in the figure legends or associated method details including replicates ($n$), trials ($N$), and statistical tests performed.

# Acknowledgements

We are grateful to Dr. Sophie Alvarez and Dr. Michael Naldrett at the Proteomics and Metabolomics Facility, Center for Biotechnology at the University of Nebraska-Lincoln for the proteomics analysis, Dr. Wen-Chao Wang at the Shanghai Profleader Biotech Co, Ltd for untargeted metabolomics analysis, and the imaging center at the School of Life Sciences, Sun Yat-Sun University for assistance with TEM and immuno-EM. We thank Prof. Vern Carruthers for the kind gift of RH*Δku80Δhxgprt* line, and Prof. Philippe Bastin for monoclonal antibodies against the epitope tag Ty (BB2). This research was supported by the National Key Research and Development Program of China (Grant Nos. 2021YFC2300800 and 2021YFC2300802) to XQZ, by the National Natural Science Foundation of China (31873009 to SL and 31772445 to DHL), and by the University Startup Package to SL.

# Additional information

## Funding

| Funder | Grant reference number | Author |
| --- | --- | --- |
| National Key Research and Development Program of China | 2021YFC2300800 | Xing-Quan Zhu |
| National Key Research and Development Program of China | 2021YFC2300802 | Xing-Quan Zhu |
| National Natural Science Foundation of China | 31873009 | Shaojun Long |

| Funder | Grant reference number | Author |
| --- | --- | --- |
| National Natural Science Foundation of China | 31772445 | De-Hua Lai |

The funders had no role in study design, data collection, and interpretation, or the decision to submit the work for publication.

## Author contributions

Hui Dong, Conceptualization, Formal analysis, Validation, Investigation, Visualization, Methodology, Writing - review and editing; Jiong Yang, Kai He, Formal analysis, Validation, Investigation, Visualization, Methodology, Writing - review and editing; Wen-Bin Zheng, Hui-Yong Ding, Rui-Bin Wu, Investigation, Visualization, Methodology; De-Hua Lai, Formal analysis, Methodology, Writing - review and editing; Jing Liu, Resources, Methodology; Kevin M Brown, Geoff Hide, Methodology, Writing - review and editing; Zhao-Rong Lun, Xing-Quan Zhu, Resources, Methodology, Writing - review and editing; Shaojun Long, Conceptualization, Resources, Formal analysis, Supervision, Funding acquisition, Validation, Visualization, Methodology, Writing - original draft, Project administration, Writing - review and editing

## Author ORCIDs

Hui Dong http://orcid.org/0000-0002-3250-871X
Kai He http://orcid.org/0000-0002-2252-4855
De-Hua Lai http://orcid.org/0000-0002-4709-1507
Geoff Hide http://orcid.org/0000-0002-3608-0175
Xing-Quan Zhu http://orcid.org/0000-0003-2530-4628
Shaojun Long http://orcid.org/0000-0002-5409-2831

## Ethics

The mouse experiments were conducted according to the guidelines and regulations issued by the Veterinary Office of the China Agricultural University (Issue No. AW11402202-2-1).

Reviewer #1 (Public Review): https://doi.org/10.7554/eLife.88866.3.sa1
Reviewer #2 (Public Review): https://doi.org/10.7554/eLife.88866.3.sa2
Author Response https://doi.org/10.7554/eLife.88866.3.sa3

---

# Additional files

## Supplementary files

• Supplementary file 1. Candidate selection and reagents of parasite lines, plasmids, and primers used in the study. (a) Selection of candidates from previous studies of HyperLOPIT AND PfACP-BirA. (b) Lines used in this study. (c) Plasmids used in this study. (d) Primers used in this study.

• Supplementary file 2. Analysis of candidates identified from our TurboID proteomics or other sources. (a) Minimally processed mass-spectrometry datasets of the APT1-TurboID fusion line and the parental line. (b) Volcano analysis of mass-spectrometry datasets from APT1-TurboID and the parental line. (c) List of known apicoplast proteins on the volcano plotting. (d) List of transmembrane domain (TMD) proteins identified by APT1-TurboID and ordered from high numbers to low numbers. (e) Merged candidate list from three datasets of hyperLOPIT, ACP-BirA, and APT1-TurboID. (f) Bioinformatic analysis of discovered transmembrane proteins in this study.

• Supplementary file 3. Identification of orthologs of apicoplast proteins in diverse taxa. (a) Identification of orthologs of novel apicoplast proteins and known proteins in chromalveolates, red algae, green algae, plants, fungi, metazoa, and bacteria. (b) Orthologs of AMT1 and AMT2 in selected species of apicomplexans.

• Supplementary file 4. Analysis of proteins lines by gas chromatography–mass spectrometry (GC–MS) and liquid chromatography–mass spectrometry (LC–MS). (a) Minimally processed datasets of metabolites detected by GC–MS and LC–MS in TIR1, AMT1-AID, AMT2-AID, and dKD lines induced by auxin for 18 hr. (b) Analysis of metabolites detected by GC–MS and LC–MS in the lines of TIR1, AMT1-AID, AMT2-AID, and dKD induced by auxin for 18 hr.

• MDAR checklist

## Data availability

The proteomic data reported in this paper have been deposited in the OMIX, China National Center for Bioinformation/Beijing Institute of Genomics, Chinese Academy of Sciences (https://ngdc.cncb. ac.cn/omix) with accession number: OMIX002059. The metabolomics raw data have been deposited in the MetaboLights (https://www.ebi.ac.uk/metabolights/) with accession number: MTBLS7060. Minimally processed data of the proteomics and metabolomics data are available in *Supplementary file 2a* and *Supplementary file 4a*, respectively. *Toxoplasma gondii* genome information can be found in ToxoDB release 53 (http://toxodb.org) and Eukaryotic pathogen, Vector & Host Information Resources can be found in VEupathDB (http://veupathdb.org).

The following datasets were generated:

| Author(s) | Year | Dataset title | Dataset URL | Database and Identifier |
| --- | --- | --- | --- | --- |
| He K | 2023 | Proximity biotnylation of proteins by APT1-TurboID | https://ngdc.cncb. ac.cn/omix/release/ OMIX002059 | OMIX, OMIX002059 |
| He K | 2024 | The Toxoplasma monocarboxylate transporters are involved in the metabolism within the apicoplast and are linked to parasite survival | https://www.ebi.ac. uk/metabolights/ MTBLS7060 | MetaboLights, MTBLS7060 |

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
