## [Editor Report · eLife assessment]

This study identifies two new transporters in the apicoplast, a non-photosynthetic organelle of apicomplexan parasites. While this is **important** work, it only partially reveals how essential these transporters are, as it does not address the metabolic function of the transporters for the parasite. Although the evidence is still **incomplete**, the results should be of interest to parasitologists and eukaryotic cell biologists.

---

## [Referee Report · Reviewer #1 (Public Review)]

The apicoplast, a non-photosynthetic vestigial chloroplast, is a key metabolic organelle for the synthesis of certain lipids in apicomplexan parasites. Although it is clear metabolite exchange between the parasite cytosol and the apicoplast must occur, very few transporters associated with the apicoplast have been identified. The current study combines data from previous studies with new data from biotin proximity labeling to identify new apicoplast resident proteins including two putative monocarboxylate transporters termed MCT1 and MCT2. The authors conduct a thorough molecular phylogenetic analysis of the newly identified apicoplast proteins and they provide compelling evidence that MCT1 and MCT2 are necessary for normal growth and plaque formation in vitro along with maintenance of the apicoplast itself. They also provide indirect evidence for a possible need for these transporters in isoprenoid biosynthesis and fatty acid biosynthesis within the apicoplast. Finally, mouse infection experiments suggest that MCT1 and MCT2 are required for normal virulence, with MCT2 completely lacking at the administered dose. Overall, this study is generally of high quality, includes extensive quantitative data, and significantly advances the field by identifying several novel apicoplast proteins together with establishing a critical role for two putative transporters in the parasite. The study, however, could be further strengthened by addressing the following aspects:

Main comments:

1. The conclusion that condition depletion of AMT1 and/or AMT2 affects apicoplast synthesis of IPP is only supported by indirect measurements (effects on host GFP uptake or trafficking, possibly due to effects on IPP dependent proteins such as rabs, and mitochondrial membrane potential, possibly due to effects on IPP dependent ubiquinone). This conclusion would be more strongly supported by directly measuring levels of IPP. If their or technical limitations that prevent direct measurement of IPP then the author should note such limitations and acknowledge in the discussion that the conclusion is based on indirect evidence.

2. The conclusion that condition depletion of AMT1 and/or AMT2 affects apicoplast synthesis of fatty acids is also poorly supported by the data. The authors do not distinguish between the lower fatty acid levels being due to reduced synthesis of fatty acids, reduced salvage of host fatty acids, or both. Indeed, the authors provide evidence that parasite endocytosis of GFP is dependent on AMT1 and AMT2. Host GFP likely enters the parasite within a membrane bound vesicle derived from the PVM. The PVM is known to harbor host-derived lipids. Hence, it is possible that some of the decrease in fatty acid levels could be due to reduced lipid salvage from the host. Experiments should be conducted to measure the synthesis and salvage of fatty acids (e.g., by metabolic flux analysis), or the authors should acknowledge that both could be affected.

---

## [Referee Report · Reviewer #2 (Public Review)]

In this study Hui Dong et al. identified and characterized two transporters of the monocarboxylate family, which they called Apcimplexan monocarboxylate 1 and 2 (AMC1/2) that the authors suggest are involved in the trafficking of metabolites in the non-photosynthetic plastid (apicoplast) of *Toxoplasma gondii* (the parasitic agent of human toxoplasmosis) to maintain parasite survival. To do so they first identified novel apicoplast transporters by conducting proximity-dependent protein labeling (TurboID), using the sole known apicoplast transporter (TgAPT) as a bait. They chose two out of the three MFS transporters identified by their screen based and protein sequence similarity and confirmed apicoplast localisation. They generated inducible knock down parasite strains for both AMC1 and AMC2, and confirmed that both transporters are essential for parasite intracellular survival, replication, and for the proper activity of key apicoplast pathways requiring pyruvate as carbon sources (FASII and MEP/DOXP). Then they show that deletion of each protein induces a loss of the apicoplast, more marked for AMC2 and affects its morphology both at its four surrounding membranes level and accumulation of material in the apicoplast stroma. The authors attempted to decipher the function of the transporters on metabolic functions of the apicoplast: (a) notably for IPP synthesis through the assessment of vesicle import allowed by IPP-based anchors, which was found to be affected in the mutants, as well as (b) apicoplast fatty acid synthesis by indirect assessment of vesicle import. However, none of them directly concluded on the actual function of the transporters. Furthermore heterologous complementation in bacterial system also failed to demonstrate the transporters' function.

However, this study is very timely, as the apicoplast holds several important metabolic functions (FASII, IPP, LPA, Heme, Fe-S clusters...), which have been revealed and studied in depth but no further respective transporter have been identified thus far. hence, new studies that could reveal how the apicoplast can acquire and deliver all the key metabolites it deals with, will have strong impact for the parasitology community as well as for the plastid evolution communities. The current study is well initiated with appropriate approaches to identify two new putatively important apicoplast transporters, and showing how essential those are for parasite intracellular development and survival. However, in its current state, this is all the study provides at this point (i.e. essential apicoplast transporters disrupting apicoplast integrity, and indirectly its major functions, FASII and IPP, as any essential apicoplast protein disruption does). The study fails to deliver further message or function regarding AMC1 and 2, and thus validate their study. Currently the manuscript just describes how AMC1/2 deletion impacts parasite survival without answering the key question about them: what do they transport. The authors yet have to perform key experiments that would reveal their metabolic function. Ideally the authors would work further and determine the function of AMC1 and 2.

---

## [Author Response]

The following is the authors’ response to the original reviews.

**Reviewer #1 (Public Review):**
The apicoplast, a non-photosynthetic vestigial chloroplast, is a key metabolic organelle for the synthesis of certain lipids in apicomplexan parasites. Although it is clear metabolite exchange between the parasite cytosol and the apicoplast must occur, very few transporters associated with the apicoplast have been identified. The current study combines data from previous studies with new data from biotin proximity labeling to identify new apicoplast resident proteins including two putative monocarboxylate transporters termed MCT1 and MCT2. The authors conduct a thorough molecular phylogenetic analysis of the newly identified apicoplast proteins and they provide compelling evidence that MCT1 and MCT2 are necessary for normal growth and plaque formation in vitro along with maintenance of the apicoplast itself. They also provide indirect evidence for a possible need for these transporters in isoprenoid biosynthesis and fatty acid biosynthesis within the apicoplast. Finally, mouse infection experiments suggest that MCT1 and MCT2 are required for normal virulence, with MCT2 completely lacking at the administered dose. Overall, this study is generally of high quality, includes extensive quantitative data, and significantly advances the field by identifying several novel apicoplast proteins together with establishing a critical role for two putative transporters in the parasite. The study, however, could be further strengthened by addressing the following aspects:

Response: We thank very much the reviewer for his/her positive evaluation of our work. To address the detailed function of the transporters, in the past three months, we have re-constructed plasmids (with codon-optimized DNA sequences of the genes) for expression of the transporters in a regular expression *E. coli* strain (BL21DE3) and in a pyruvate import knockout *E. coli* strain (a gift from Prof. Kirsten Jung), to examine the transport capability in vitro. And, we have also re-constructed a new plasmid containing a new leading peptide for targeting the pyruvate sensor PyronicSF to the apicoplast in the parasite, to probe the possible substrate pyruvate. However, we did not successfully observe expression of the transporters in the above *E. coli* strains, and we were unable to target the sensor to the correct localization (the apicoplast) in the parasite. As a result, all efforts have led the study to the current version of manuscript on the functional identification of transporters. We will keep working on this aspect, attempting to dissect out the exact transport function of the transporters in the future. In the current manuscript, we have discussed the limitations of our study in the last part of the manuscript.

Main comments1. The conclusion that condition depletion of AMT1 and/or AMT2 affects apicoplast synthesis of IPP is only supported by indirect measurements (effects on host GFP uptake or trafficking, possibly due to effects on IPP dependent proteins such as rabs, and mitochondrial membrane potential, possibly due to effects on IPP dependent ubiquinone). This conclusion would be more strongly supported by directly measuring levels of IPP. If there are technical limitations that prevent direct measurement of IPP then the author should note such limitations and acknowledge in the discussion that the conclusion is based on indirect evidence.

Response: We thank the reviewer very much for the suggestions. We have tried to establish the measurement of IPP using a commercial company in recent months, yet we have not been successful in making the assay work. Considering the problem of indirect evidence, we have discussed this limitation in the discussion.

1. The conclusion that condition depletion of AMT1 and/or AMT2 affects apicoplast synthesis of fatty acids is also poorly supported by the data. The authors do not distinguish between the lower fatty acid levels being due to reduced synthesis of fatty acids, reduced salvage of host fatty acids, or both. Indeed, the authors provide evidence that parasite endocytosis of GFP is dependent on AMT1 and AMT2. Host GFP likely enters the parasite within a membrane bound vesicle derived from the PVM. The PVM is known to harbor host-derived lipids. Hence, it is possible that some of the decrease in fatty acid levels could be due to reduced lipid salvage from the host. Experiments should be conducted to measure the synthesis and salvage of fatty acids (e.g., by metabolic flux analysis), or the authors should acknowledge that both could be affected.

Response: We thank the reviewer very much for comments and suggestions. We partially agree with the comments that the depletion of transporters could affect lipids scavenged from the host cells, as endocytic vesicles are indeed derived from the parasite plasma membrane at the micropore and potentially from the host cell endo-membrane system, as demonstrated with the micropore endocytosis in our previous study (pmid: 36813769). Our latest study has addressed this by showing that the endocytic trafficking of GFP vesicles is regulated by prenylation of proteins (e.g. Rab1B and YKT6.1), depletion of which resulted in diffusion of GFP vesicles, but not disappearance of GFP vesicles in the parasites (pmid: 37548452), indicating that the vesicles (containing lipids) enter the parasites. In the current manuscript, the percentage of parasites containing GFP foci was significantly reduced in AMT1/AMT2-depleted parasites, and instead, parasites containing GFP diffusion appeared and the percentage was almost equal to the reduced level of parasites with GFP foci. These results suggested that endocytic vesicles (e.g. GFP vesicles) were continuously generated by the micropore in the parasites depleted with AMT1/AMT2, and that the vesicle trafficking was regulated by proteins modified by IPP derivatives that were derived from the apicoplast. Based on these observations, we considered that lipids in endocytic vesicles should not contribute to the reduced level of fatty acids and other lipids in parasites depleted with AMT1/AMT2. We have added in a short discussion concerning the fatty acids and lipids reduced in the parasites.

**Reviewer #2 (Public Review):**
In this study Hui Dong et al. identified and characterized two transporters of the monocarboxylate family, which they called Apcimplexan monocarboxylate 1 and 2 (AMC1/2) that the authors suggest are involved in the trafficking of metabolites in the non-photosynthetic plastid (apicoplast) of *Toxoplasma gondii* (the parasitic agent of human toxoplasmosis) to maintain parasite survival. To do so they first identified novel apicoplast transporters by conducting proximity-dependent protein labeling (TurboID), using the sole known apicoplast transporter (TgAPT) as a bait. They chose two out of the three MFS transporters identified by their screen based and protein sequence similarity and confirmed apicoplast localisation. They generated inducible knock down parasite strains for both AMC1 and AMC2, and confirmed that both transporters are essential for parasite intracellular survival, replication, and for the proper activity of key apicoplast pathways requiring pyruvate as carbon sources (FASII and MEP/DOXP). Then they show that deletion of each protein induces a loss of the apicoplast, more marked for AMC2 and affects its morphology both at its four surrounding membranes level and accumulation of material in the apicoplast stroma. This study is very timely, as the apicoplast holds several important metabolic functions (FASII, IPP, LPA, Heme, Fe-S clusters...), which have been revealed and studied in depth but no further respective transporter have been identified thus far. hence, new studies that could reveal how the apicoplast can acquire and deliver all the key metabolites it deals with, will have strong impact for the parasitology community as well as for the plastid evolution communities. The current study is well initiated with appropriate approaches to identify two new putatively important apicoplast transporters, and showing how essential those are for parasite intracellular development and survival. However, in its current state, this is all the study provides at this point (i.e. essential apicoplast transporters disrupting apicoplast integrity, and indirectly its major functions, FASII and IPP, as any essential apicoplast protein disruption does). The study fails to deliver further message or function regarding AMC1 and 2, and thus validate their study. Currently, the manuscript just describes how AMC1/2 deletion impacts parasite survival without answering the key question about them: what do they transport? The authors yet have to perform key experiments that would reveal their metabolic function. I would thus recommend the authors work further and determine the function of AMC1 and 2.

Response: We thank very much the reviewer for his/her positive evaluation of our work. To address the detailed function of the transporters, in the past three months, we have re-constructed plasmids (with codon-optimized DNA sequences of the genes) for expression of the transporters in a regular expression *E. coli* strain (BL21DE3) and in a pyruvate import knockout *E. coli* strain (a gift from Prof. Kirsten Jung), to examine the transport capability in vitro. And, we have re-constructed a new plasmid containing a new leading peptide for targeting the pyruvate sensor PyronicSF to the apicoplast in the parasite, to probe the possible substrate pyruvate. However, we were unable to successfully observe expression of the transporters in the above *E. coli* strains, and we were unable to target the sensor to the correct localization (the apicoplast) in the parasite. As a result, all these efforts have led the study to the current version of manuscript on the functional identification of transporters. We will keep working on this aspect, attempting to dissect out the exact transport function of the transporters in the near future. In this current manuscript, we have discussed the limitations of our study in the last part of the manuscript.

**Reviewer #1 (Recommendations For The Authors):**
Minor commentsLine 35: ...appears to have evolved...Line 67: remove first commaLine 105: thereafter or therefore?Line 130: define ACPLine 131: define TMD

Response: We thank very much the reviewer for the suggestions, and we have revised the points in the current manuscript.

Figure 1: more information on APT1 would be helpful for readers to interpret the results from turboID e.g., consider showing an illustration showing, according to Karnataki et al 2007 that APT1 likely occupies all 4 membranes of the apicoplast. Also, according to DeRocher et al 2012, APT1 N-term and C-term are both cytosolically exposed, at least in the outermost membrane. The orientation in the other membranes is not known.

Response: We thank very much the reviewer for the suggestions. We analyzed the localization information of APT1 in *T. gondii*, based on the studies as the reviewer proposed (Karnataki, et al., 2007; DeRocher et al., 2012). The HA tag at the C-terminus of APT1 was distributed at the four membranes of the apicoplast, indicating that the topology of APT1 might be difficult to be defined at the membranes. Considering this information, we felt hesitant to clearly describe the topology in a schematic diagram about the protein APT1. Nevertheless, the TurboID tagging at the C-terminus of APT1 was an excellent model for identification of potential transporters localized at membranes of the apicoplast. We have put more information about the topology of APT1 in the manuscript, thus providing a better understanding of the proteomic results.

Figure 2: add a space between "T." and "gondii"Figure 2: remove period between "Fitness" and "scores"Figure 2: different fonts are used within the figure. Consider using only one font such as arial. Same for Figure 4.Figure 2: "Fitness scores" is not bold in panel A but is bold in panel B.

Response: We thank very much the reviewer for the suggestions. We have revised the points in the current version of the manuscript.

Line 187: superscript -7Line 249: Caution should be used in interpreting two bands as being a precursor and mature product without additional experiments to establish such a relationship. Consider using the term "might" rather than "appear to". The presence of multiple bands could be due to phenomena other than proteolytic processing e.g., alternative splicing, alternative initiator codons, etc.

Response: We thank very much the reviewer for the suggestions. We have revised the sentences in the current version of manuscript.

Line 291: define IPPFigure 3E. The data points for KD strains appear to be positioned above the zero value on the y-axis. Is this correct?

Response: We thank very much the reviewer for the suggestions. We have rechecked the figure and replaced it with the correct one.

Figure 3 G/H legend. Please describe what a single data point represents e.g., the average of one field of view, the average of a certain number of fields of view, or something else? Are the data combined from three experiments or from a representative experiment?

Response: We thank very much the reviewer for the suggestions. Three independent experiments were performed with at least three replicates. At least 150 vacuoles were scored in each replicate, thus resulting in at least 9 data points in total. The data points were shown with the results from each replicate.

Line 325: define MEP and explain how it is connected to IPP

Response: We thank very much the reviewer for the suggestions. We have provided the information in the current version of the manuscript.

Lines 351-355: The authors refer to Figure 4D to support this statement, but presumably they mean 4E. Also, the authors use the terms C14, C16, and C18. They should more precisely use the terms myristic acid, palmitoleic acid, and trans_oleic acid if this is what they are referring to. Finally, the authors should determine if there is a statistically significant difference between levels of these fatty acids between AMT1 KD and AMT2 KD. If not, they should suggest there is an overall trend toward lower levels of these fatty acids in AMT2 KD parasites compared to AMT1 KD parasites.

Response: We thank very much the reviewer for the suggestions. We have revised the information in the current version of the manuscript.

Lines 363-364: The basis of this comment is unclear. Please clarify.Lines 369-370: the authors have not shown that the observed lower levels of fatty acids are due to synthesis, as noted above

Response: We thank very much the reviewer for the suggestions. We have accordingly revised the information in the current version of the manuscript.

Line 383: Should be Figure S6DLine 386: An entire section of the results is used to describe data that are entirely in a supplemental figure. Consider moving this data to a main figure.

Response: We thank very much the reviewer for the suggestions. We have transferred the data to the main figure in the current version of the manuscript.

Line 391: Consider using the term virulence instead of growth since now experiments were performed to specifically assess parasite growth in the infected mice.

Response: We thank very much the reviewer for the suggestions. We have revised the terms in the Results section.

Line 427: Perhaps the authors mean "...strong growth defect..." or ...strong growth impairment..."Line 460-461: This statement is unclear. Please explain how strong backgrounds in proteomics have made it difficult to identify apicoplast transporters. Because they are low abundance? Because they are membrane proteins?

Response: We thank very much the reviewer for the suggestions. We have revised the corresponding sentences in the current version. The strong backgrounds in the proteomics resulted from the high activity and nonspecific labeling of biotin ligase fused with the apicoplast proteins.

518-521: It would be helpful for non-specialists if the authors explained how pyruvate is connected to IPP biosynthesis.523: delete period after "Escherichia"548-549: "We observed similar decreases in level of the MEP biosynthesis activity upon depletion of AMT1 and AMT2..." Reword this since no experiments were done to measure MEP biosynthesis activity.

Response: We thank very much the reviewer for the suggestions. We have accordingly revised the relevant sentences in the manuscript.

**Reviewer #2 (Recommendations For The Authors):**
Major points:The metabolomic data on fatty acid synthesis and isoprenoid levels is relevant but cannot inform about the function of the transporter, since any protein causing loss of the apicoplast would behave in such a manner, i.e. block the apicoplast pathways.

Response: We thank very much the reviewer for the comment. We agree with this comment. We have thus discussed these points in a subsection in the Discussion, pointing out some of the limitations in the study.

Currently, the manuscript fails to directly prove what AMC1 and AMC2 transports, potentially pyruvate as suggested to putatively fuel FASII and MEP/DOXP. Further experimental approaches using exogenous complementation and/or metabolomic analyses using stable isotope labelling (for example) should potentially bring light to the putative functions of AMC1/2.

Response: We thank very much the reviewer for the comments. As described above, we attempted several approaches to find out the substrates that the AMT1 and AMT2 transports. However, we could not successfully express the proteins in *E. coli* strains, and we did not generate a *T. gondii* strain that a pyruvate sensor was properly targeted to the apicoplast. At the end of the Discussion, we have a subsection that discusses the limitations of this study. We hope that our future approaches will be able to tackle these difficulties on the substrate identification.

Furthermore, the authors have not considered other pathways of interest, like heme or lysophosphatidic acid (LPA)n synthesis, which are two other key pathway, which may be related to AMC1/2 function.Those proposed experiments represent an important body of work, required to bring light to their metabolic functions.

Response: We thank very much the reviewer for the comments. We thought about that, but we finally decided to mainly discuss two of the pathways that the transporters might participate in, since the transporters contain specific domains on the proteins sequences that potentially are associated with pyruvate.

Further, the authors might have partially missed some referencing and data about the apicoplast in their introduction (and potentially to address other facets of the apicoplast metabolic functions/capacities in regards to AMC1/2 function): the introduction referencing and explanations are somehow not fully exact/precise for the part of the apicoplast and its pathway: references about the apicoplast, discovery and origin are not citing the original work (that should be Wilson et al. 1996, McFadden et al. 1996, Kohler et al. 1997,), same for the discovery of FASII and MEP./DOXP (Waller 1998, Jomaa et al...). The introduction (and the study?) lacks information about other key functions of the apicoplast: heme synthesis, lysophosphatidic acid synthesis (using FASII products). The explanations about the roles of FASII/DOXP are partial and not fully citing important references: Krishnan et al. 2020, and Amiar et al. 2020 are also key to understanding how the role of FASII is metabolically flexible depending on nutrient content. A whole part on the fact that FASII is not only dispensible but can also become essential under metabolic adaptations conditions, are missing (Botté et al. 2013, Amiar et al. 2020, Primo et al. 2021). These novel important facets of parasite biology should be mentioned as well as directly linked to the author's topic. This is more minor but could bring new ideas to the authors.

Response: We thank very much the reviewer for the suggestions. We have revised the relevant part in the introduction.

We are grateful for the suggestions to improve the manuscript.